# Density-Aware Translation of Spurious Correlations in Zero-Shot VLMs

**Afsaneh Hasanebrahimi** [1]  **Hanxun Huang** [1]  **Christopher Leckie** [1]  **Sarah Erfani** [1]

## Abstract

Vision-Language models (VLMs), such as CLIP, achieve powerful zero-shot classification. However, their predictions remain sensitive to spurious correlations, where contextual cues dominate over semantic content. Earlier solutions typically rely on fine-tuning or prompt engineering, which either undermine the advantages of pre-trained models or are prone to hallucination. In this work, we propose **Density-Aware Translation** (DAT) that refines image-text similarity scores using a local geometric density term derived from group reference sets. Our approach is motivated by the phenomenon that CLIP embeddings exhibit a modality gap and lie on an anisotropic shell in the feature space: common patterns cluster near the mean, while rare patterns are pushed outward. This geometry creates uneven alignment, where spurious correlations are amplified while semantically meaningful but rare cues are marginalised. To address this, we employ a relative measure to rescale similarities based on embedding density, suppressing overconfident scores in diffuse regions while preserving dense, semantically consistent matches. Experimental results on benchmark datasets demonstrate consistent improvements in worst-group and average accuracy, highlighting density-aware translation as a simple and effective calibration mechanism for reliable zero-shot classification using multimodal models.

## 1. Introduction

Vision-Language models (VLMs), such as CLIP (Radford et al., 2021), have advanced multimodal learning by aligning image and text embeddings in a shared latent space, enabling strong zero-shot capabilities that generalise to unseen classes without requiring additional training (Chen et al., 2024). This capability makes them widely used for tasks such as classification (Qian & Hu, 2024), retrieval (Sain et al., 2023), and reasoning (Subramanian et al., 2022) in different domains.

Despite their remarkable success, VLMs are still susceptible to spurious correlations (Bommasani, 2021; Chuang et al., 2023; Varma et al., 2024), where predictions rely on frequent but semantically irrelevant cues rather than meaningful content. Levi & Gilboa (2025) show that frequently occurring concepts tend to align more closely with the modality mean vector, exhibiting higher conformity, which reflects semantic blurring. This geometric bias makes the model over-rely on such frequent but uninformative patterns, while down-weighting rarer yet more informative signals, ultimately compromising robustness across groups. For instance, spurious correlations arise in chest X-ray diagnostics, where models trained to detect pneumonia have been shown to rely on hospital-specific markers instead of true lung pathology (Zech et al., 2018; DeGrave et al., 2021).

Existing approaches to mitigating spurious correlations in VLMs fall into three groups. (i) Fine-tuning and adaptors (Zhang & Ré, 2022; Goyal et al., 2023; Varma et al., 2024) rely on bias-aware objectives but require labelled supervision, undermining zero-shot generalisation. Wu et al. (2023) also propose a supervised, concept-aware correction using gradient-based concept discovery with the white-box model access. (ii) Text-side methods (Chuang et al., 2023; Trager et al., 2023; An et al., 2024) edit prompts or projections, but risk cross-modal misalignment and often depend on domain expertise or LLMs, which can be unreliable and inconsistent (Abbasi et al., 2025; Huang et al., 2025; Molahasani et al., 2025). (iii) Multimodal embedding adjustments (Adila et al., 2024; Lu et al., 2025) alter image features with text guidance, but linear projections distort geometry and translation-based methods require dataset-specific scaling, calibrated by training data.

In this work, we revisit the geometric limitations of VLMs such as CLIP, by examining how their similarity scores neglect the anisotropic, ellipsoidal structure of the embedding space. Prior studies have shown that CLIP embeddings are

[1]School of Computing and Information Systems, The University of Melbourne, Victoria, Australia. Correspondence to: Afsaneh Hasanebrahimi <a.hasanebrahimi@unimelb.edu.au>.

*Proceedings of the 43rd International Conference on Machine Learning*, Seoul, South Korea. PMLR 306, 2026. Copyright 2026 by the author(s).

unevenly distributed, with dense and sparse regions emerging along different directions (Liang et al., 2022; Levi & Gilboa, 2025). Building on this observation, we propose the **Density-Aware Translation** (DAT) that incorporates *local geometric information* into similarity computation. In particular, we quantify local geometry via a relative density ratio, estimated from reference sets constructed through sampling to capture variations across both labels and spurious attributes. This allows us to down-weight spurious similarities that occur when a sample aligns with the wrong prompt, while preserving similarity for samples that lie in dense, representative regions of their correct group. Importantly, our approach operates fully in the zero-shot regime, without access to model parameters, and only relies on a small, balanced reference set to capture group-level density characteristics.

From a theoretical perspective, we model group distributions using the Kent (Fisher-Bingham) distribution (Kent, 1982; Mardia & Jupp, 2009), and show that raw CLIP scores systematically deviate from Bayes-optimal decision boundaries. Our DAT provably reinstates anisotropy-sensitive log-likelihood terms that cosine similarity ignores, aligning the discriminant with Bayes-optimal scoring. We evaluate DAT on standard spurious correlation benchmarks and demonstrate consistent improvements in both worst-group and average accuracy, while preserving the flexibility of zero-shot inference. These results underscore the importance of explicitly modelling anisotropy and local density in multimodal embeddings to achieve robust classification.

The main contributions can be summarised as follows:

- We introduce DAT, a zero-shot mechanism that rescales image–text similarities using group reference densities, requiring no fine-tuning, no prompt engineering, and no access to spurious attribute labels at test time.

- We provide a theoretical analysis showing that DAT corrects cosine's bias under anisotropic embeddings, reinstating missing log-likelihood terms and aligning with Bayes-optimal decision rules.

- Through experiments on different benchmarks and across multiple VLMs, we demonstrate consistent improvements in different metrics.

## 2. Related Work

**Debiasing VLMs via Fine-tuning.** Several recent works adapt CLIP and related VLMs through bias-aware fine-tuning objectives. Yang et al. (2023) introduces a multimodal contrastive loss that explicitly separates spurious attributes from class-defining features by encoding spurious cues in language. Varma et al. (2024) identifies spurious correlations at the region level via clustering, and then applies

a region-aware loss that suppresses spurious regions while emphasising causal ones. Zhang & Ré (2022) develops contrastive adaptors that not only align sample embeddings with their class prototypes but also bring same-class samples closer together, improving group robustness with minimal parameters. Zhang et al. (2024) propose a prompt tuning method that disentangles causal and spurious features by decoupling alignment into two contrastive phases. Pang et al. (2024) instead leverages vectorised group attributes to explicitly debias image representations under sub-population shifts. Dehdashtian et al. (2024) frame CLIP debiasing in a reproducing kernel Hilbert space and employ a statistical dependence measure to decorrelate representations from spurious attributes.

**Zero-shot Debiasing in VLMs.** Some methods aim to mitigate spurious correlations through embedding-based debiasing without retraining, thereby preserving the zero-shot capability of VLMs. Ideal-Prompt (Trager et al., 2023) constructs an ideal text representation by combining basis vectors in the embedding space. Perception CLIP (An et al., 2024) is a two-stage procedure that first infers contextual attributes (e.g., background) and then conditions object classification on them. Orth-Cali (Chuang et al., 2023) project text embeddings onto subspaces orthogonal to spurious directions, showing that text-only calibration suffices for robust classifiers and fair generative models. Ge et al. (2023) improves zero-shot accuracy by detecting potentially misclassified images via prediction consistency and augmenting text prompts with hierarchical label information from WordNet. Adila et al. (2024) proposes ROBOSHOT that leverages large language models to extract spurious-related insights from task descriptions and applies linear projection to suppress harmful and enhance beneficial embedding components, though it inherits fragility from LLM-generated cues. Most recently, Lu et al. (2025) presents TIE, a zero-shot framework that translates image embeddings along directions guided by spurious text prompts, reducing reliance on shortcuts while preserving distributional structure. In TIE, it is assumed that the spurious labels of the test images are available. To relax this assumption, TIE* is introduced, which operates without access to spurious labels.

## 3. Density-Aware Translation

In this section, we begin with the formal problem statement in Section 3.1, followed by the motivation in Section 3.2, the proposed DAT in Section 3.3, and finally the theoretical analysis in Section 3.4.

### 3.1. Problem Statement

We study the group robustness setting (Sagawa et al., 2020) in the context of zero-shot classification. We adopt the stan-

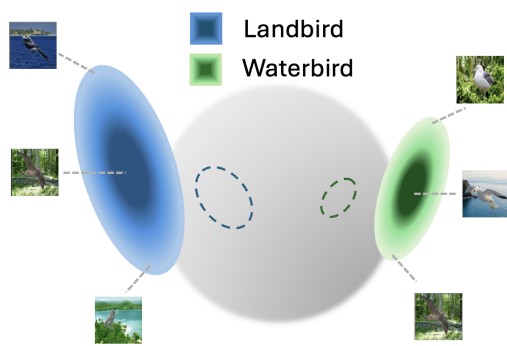

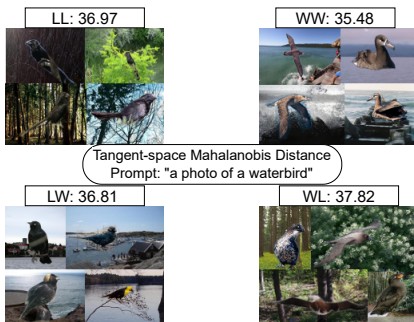

*(a)* Ellipsoidal shape of embedding in CLIP.

*(b)* Distance inconsistency in CLIP.

*Figure 1.* Motivation for DAT. (a) CLIP embeddings exhibit anisotropic ellipsoidal structure, where frequent, spuriously correlated samples cluster near the mean, while rare but semantically meaningful samples lie in sparser regions. (b) Tangent-space Mahalanobis Distance for group embeddings shows that spuriously aligned LW has a lower distance, on average, to the waterbird prompt than the true WL group. Abbreviations: *LL*: landbird with land background, *WL*: waterbird with land background, *LW*: landbird with water background, *WW*: waterbird with water background.

dard definition of zero-shot, where a model is expected to correctly classify samples from unseen classes at test time (Wang et al., 2019). A detailed discussion of DAT's conformity to this setting, in relation to prior work on spurious correlations, is provided in Appendix B.3.

Let $(x, y, a) \in \mathcal{X} \times \mathcal{Y} \times \mathcal{A}$ denote an input image $x$, with class label $y \in \mathcal{Y}$, and a spurious attribute $a \in \mathcal{A}$ (e.g., background or context). We denote $|\mathcal{Y}| = K$ for the number of classes and $|\mathcal{A}| = M$ for the number of spurious attributes. Each group is defined as a pair $(y, a)$,

$$g_{y,a} \in \mathcal{G} = \mathcal{Y} \times \mathcal{A}, \quad |\mathcal{G}| = K \cdot M,$$

so that robustness is evaluated at the group level. For the zero-shot model, we denote $\phi_I(.)$ as the frozen image encoder and $\phi_T(.)$ as the frozen text encoder.

Following Lu et al. (2025), we assume access to text prompts describing both labels and spurious attributes. For each label $y \in \mathcal{Y}$, we define a class-only prompt $t_y \in \mathcal{T}^{\mathcal{Y}}$, e.g., "a photo of a y". For each spurious attribute $a \in \mathcal{A}$, we define an attribute prompt $t_a \in \mathcal{T}^{\mathcal{A}}$, e.g., "a photo of a a". Finally, to represent groups, we construct concatenated prompts as group prompts $t_{y,a} \in \mathcal{T}^{\mathcal{Y},\mathcal{A}}$, e.g., "a photo of a y with a". To capture group geometry, we consider group samples from the training or validation set as $\{x_{y,a}^{(h)}\}_{h=1}^{N_{y,a}}$, where $N_{y,a}$ denotes the number of available samples in group $(y, a)$. When the spurious attribute $a$ is not explicitly provided, we employ DAT*, which infers group membership in a zero-shot manner via attribute prompts:

$$\hat{a} = \arg\max_{a \in \mathcal{A}} \langle \phi_I(x), \phi_T(t_a) \rangle. \quad (1)$$

The corresponding image embeddings are then obtained as

$$\text{DAT}: z_{y,a}^{(h)} = \phi_I(x_{y,a}^{(h)}), \qquad \text{DAT*}: z_{y,\hat{a}}^{(h)} = \phi_I(x_{y,\hat{a}}^{(h)}).$$

The reference samples are used only to estimate non-parametric group-level geometry in the frozen embedding space; they are not used to update model parameters or tune the VLM. Thus, DAT preserves the zero-shot inference setting. When spurious labels are unavailable, DAT* replaces explicit group annotations with zero-shot attribute inference.

### 3.2. Understanding the Geometric Bias

As illustrated in Figure 1a, CLIP embeddings are distributed anisotropically on ellipsoidal shells: frequent concepts (often correlated with spurious attributes) cluster closer to the mean, while rare but semantically meaningful concepts lie in sparser, peripheral regions (Levi & Gilboa, 2025). For example, in the Waterbirds dataset (Sagawa et al., 2020), "waterbird on water" is much more common than "waterbird on land." CLIP's embedding geometry places these spuriously correlated samples near the centre, while rare but informative cases occupy the periphery.

This geometric bias creates two key challenges for zero-shot classification. First, scoring against a single prompt per class means rare but valid samples can receive lower similarity than spuriously aligned ones, sometimes matching spurious prompts more strongly than their true class. Second, CLIP often overemphasises the largest object and the first prompt token while down-weighting smaller but critical details (Abbasi et al., 2025), letting background or context cues dominate and limiting the value of prompt augmentation alone.

We analyse group misalignment using the Tangent–Space Mahalanobis Distance (TMD) on the unit sphere (Pennec, 2006). Let all embeddings be $\ell_2$–normalised, $\hat{\mu}_{y,a}$ be group Fréchet mean on the unit hypersphere $\mathbb{S}^{d-1}$ and $\text{Log}_{\hat{\mu}_{y,a}}$ be the Riemannian logarithm map that projects points from the

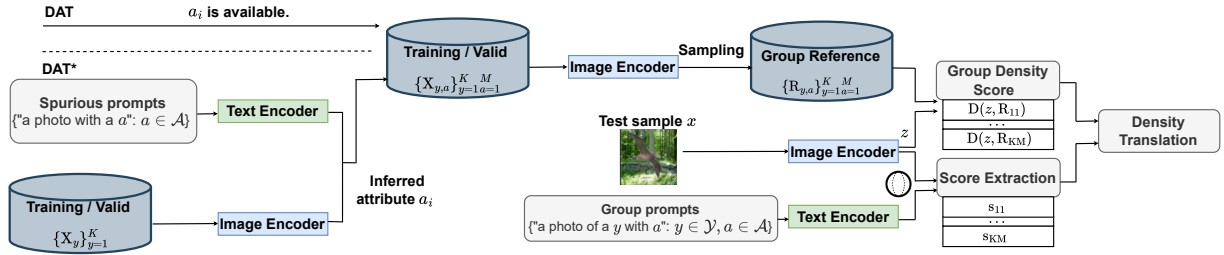

*Figure 2.* Pipeline of DAT/DAT*: DAT constructs group reference sets, encodes test images and prompts, estimates local density, and rescales similarities. DAT* follows the same pipeline but infers group attributes automatically, without explicit annotations.

sphere to the tangent space at $\hat{\mu}_{y,a}$. Let $z_{y,a}^{(h)} \in \mathbb{S}^{d-1}$ be the $h$-th image embedding in group $(y, a)$, and $w_{y,a} \in \mathbb{S}^{d-1}$ be the normalised text embedding. Then we define:

$$\delta_{y,a}^{(h)} = \mathrm{Log}_{\hat{\mu}_{y,a}}\big(z_{y,a}^{(h)}\big), \quad \delta_{w,y,a} = \mathrm{Log}_{\hat{\mu}_{y,a}}\big(w_{y,a}\big),$$

$$\Sigma_{y,a} = \frac{1}{N_{y,a}} \sum_{h=1}^{N_{y,a}} \delta_{y,a}^{(h)} \delta_{y,a}^{(h)\top},$$

$$\mathrm{TMD}_{y,a} = \sqrt{\delta_{w,y,a}^{\top}\big(\Sigma_{y,a}\big)^{-1}\delta_{w,y,a}},$$

where $\Sigma_{y,a}$ denotes the empirical covariance of tangent vectors within group $(y, a)$, and $\delta_{w,y,a}$ represents the projection of the text embedding into the tangent space at the group mean. As shown in Figure 1b, using Waterbirds with CLIP ViT-L/14, the spuriously aligned group LW (landbird on water) has a smaller TMD to the "a photo of a waterbird" prompt than the true WL (waterbird on land) group. This indicates that background correlation (water) dominates alignment, biasing CLIP toward spuriously aligned groups and away from semantically correct ones.

### 3.3. Debiasing via Density-Aware Translation

As shown in Figure 2, we first construct reference sets per group using a sampling procedure with an equal budget $n$ per group. We then apply deterministic feature-space exemplar herding (Rebuffi et al., 2017) (Appendix B.2), which greedily selects embeddings whose running average matches the group mean. This yields a compact set of representative exemplars for each $(y, a)$, providing stable local neighbourhoods for density estimation. Following Levi & Gilboa (2025), frequent (common) samples tend to lie closer to the centre of the embedding distribution, so sampling towards the mean allows us to capture these typical examples and compute a fair density metric for each group. For each group $(y, a)$ we construct a reference set of size $n$:

$$\{z_{y,a}^{(h)}\}_{h=1}^{N_{y,a}} \xrightarrow{\text{Sampling}} R_{y,a} = \{z_{y,a}^{(h)}\}_{h=1}^{n}$$

Bias annotations (e.g., background or gender) can optionally be used, but are not required. In cases without explicit group labels, we have **DAT***, which employs an auxiliary zero-shot classifier to infer group membership using Equation 1.

**Density Ratio Estimation.** For each query image $x \in \mathcal{X}_{\text{test}}$, we then compute its embedding $z = \phi_I(x)$. The local density of $z$ relative to $R_{y,a}$ is estimated with the simplified local outlier factor (SLOF) (Schubert et al., 2014) as our default proxy:

$$D_{y,a}(z) = \mathrm{SLOF}(z; R_{y,a}) = \frac{1}{k} \sum_{z_o \in \mathrm{NN}_k(z)} \frac{\text{k-dist}(z)}{\text{k-dist}(z_o)}.$$

$\mathrm{NN}_k(z)$ denotes the $k$-nearest neighbours of $z$ in $R_{y,a}$ and k-dist$(\cdot)$ is the distance to the $k$-th neighbour. A larger SLOF value indicates that $z$ lies in a sparser region, i.e., it is less representative of the group. We use SLOF due to its simplicity. Other relative density measures are also applicable. We analyze the effect of different density estimation methods in Appendix B.6.

**Density Translation.** Given group prompts $t_{y,a}$ and class-only prompts $t_y$, we compute the raw similarities as

$$s_{y,a}(x) = \langle \phi_I(x), \phi_T(t_{y,a}) \rangle, \qquad s_y(x) = \langle \phi_I(x), \phi_T(t_y) \rangle.$$

We then translate group scores using local density:

$$\tilde{s}_{y,a}(x) = \frac{s_{y,a}(x)}{(D_{y,a}(z) + \varepsilon)^{\lambda}}, \tag{2}$$

where $\lambda > 0$ controls the translation strength and small $\varepsilon > 0$ ensures numerical stability. DAT adjusts raw similarities by reducing scores for samples that appear highly similar to a group prompt but lie far from the core of a group's image embeddings, while preserving scores for samples close to their true group.

**Aggregation and Prediction.** In addition to group-specific scores $\{\tilde{s}_{y,a}(x)\}_{a \in \mathcal{A}}$, we define a class-marginal (attribute-averaged) score:

$$\tilde{s}_{y,\mathrm{Avg}}(x) = \frac{1}{M+1}\Big( \sum_{a \in \mathcal{A}} \tilde{s}_{y,a}(x) + s_y(x) \Big). \tag{3}$$

Final predictions are made by maximising over group-corrected and averaged scores:

$$\hat{y}_q \;=\; \arg\max_{y \in \mathcal{Y}} \Big\{ \max_{a \in \mathcal{A}} \tilde{s}_{y,a}(x),\; \tilde{s}_{y,\mathrm{Avg}}(x) \Big\}. \quad (4)$$

The full procedure of DAT/DAT* is outlined in Algorithm 1 and Algorithm 2 in Appendix B.1. Appendix C further illustrates how density shapes predictions, showing that SLOF separates rare samples from spuriously frequent ones and that DAT enlarges the score margin over the baseline.

### 3.4. Theoretical Analysis

We first show that raw cosine similarity is systematically biased in anisotropic embedding geometries. We then assume a standard log-density fidelity link for SLOF, and prove that DAT reinstates the anisotropy-sensitive terms missing from pure cosine scoring.

**Anisotropy Effect on Cosine Similarity.** Due to the anisotropic nature of CLIP embeddings, a query embedding $z \in \mathbb{S}^{d-1}$ can exhibit elliptical concentration, which is naturally captured by the Kent (Fisher–Bingham) distribution (Kent, 1982; Mardia & Jupp, 2009).

With parameters $(\kappa, \beta, \Gamma)$ and orthonormal frame $\Gamma = (\gamma_1, \gamma_2, \gamma_3, \ldots)$, its density is

$$p(z \mid \kappa, \beta, \Gamma) = c_d(\kappa, \beta) \exp\big(\kappa\, \gamma_1^\top z\big)$$
$$\times \exp\big(\beta[(\gamma_2^\top z)^2 - (\gamma_3^\top z)^2]\big), \quad z \in \mathbb{S}^{d-1}.$$

Here, $\kappa \geq 0$ controls axial concentration toward $\gamma_1$, and $\beta$ controls ellipticity (anisotropy) in the $(\gamma_2, \gamma_3)$-plane. The Bayes log-density is therefore

$$\log p(z) = \kappa\, \gamma_1^\top z + \beta\big[(\gamma_2^\top z)^2 - (\gamma_3^\top z)^2\big] - \log c_d(\kappa, \beta). \quad (5)$$

Cosine scoring with text direction $w$ uses only the axial projection $w^\top z$. In particular, if we take $w = \gamma_1$ (the Kent mean axis), cosine recovers the linear term $\kappa\, \gamma_1^\top z$ but ignores the quadratic anisotropy term $\beta\big[(\gamma_2^\top z)^2 - (\gamma_3^\top z)^2\big]$.

**Proposition 3.1.** *Let $p$ be $\mathrm{Kent}(\kappa, \beta, \Gamma)$ with $\beta \neq 0$ and set the cosine direction $w_y = \gamma_1$. Then there exist $z_+, z_- \in \mathbb{S}^{d-1}$ with $w_y^\top z_+ = w_y^\top z_-$ but $\log p(z_+) > \log p(z_-)$. Hence, ranking by cosine similarity can disagree with Bayes ranking $\log p(z)$.*

This shows that cosine similarity systematically overlooks anisotropy effects, potentially misranking rare but semantically important samples.

**Assumption 3.2** (log-SLOF fidelity). *There exist constants $\alpha > 0$, $\eta \in \mathbb{R}$, and a bounded error $\epsilon_{y,a}(z)$ with $|\epsilon_{y,a}(z)| \leq c$ such that*

$$\log D_{y,a}(z) \;=\; \alpha\big(-\log p_{y,a}(z)\big) \;+\; \eta \;+\; \epsilon_{y,a}(z),$$

*where $p_{y,a}(z)$ is the group density.*

This assumption formalises the link between $k$NN distance statistics and negative log-density (Loftsgaarden & Quesenberry, 1965; Biau & Devroye, 2015). LOF/SLOF are widely used density proxies that satisfy such a relation up to bounded error (Breunig et al., 2000; Zimek et al., 2012).

**DAT Margin.** Given group $(y, a)$, let $s_{y,a}(x) = \langle \phi_I(x), \phi_T(t_{y,a}) \rangle$ denote the raw similarity (cosine between image and text embeddings), and let $D_{y,a}(z)$ be a local sparsity proxy (SLOF). For the purpose of theoretical analysis, we work in the logit space, defining

$$\ell_{y,a}(x) := \tau\, w_{y,a}^\top z,$$

where $w_{y,a}$ is the normalised text embedding and $\tau > 0$ is a temperature. In logit space, Eq. 2 corresponds to subtracting $\lambda \log D$ from the raw logit $\ell$. We then define the DAT margin as

$$m_{y,a}(z) \;:=\; \ell_{y,a}(z) \;-\; \lambda \log D_{y,a}(z).$$

**Theorem 3.3** (Local Bayes alignment). *Under Assumption 3.2,*

$$m_{y,a}(z) \;=\; \tau\, w_{y,a}^\top z \;+\; \alpha\lambda \log p_{y,a}(z) \;+\; r_{y,a}(z),$$

$$r_{y,a}(z) := -\lambda\eta - \lambda\, \epsilon_{y,a}(z),$$

*so that $|r_{y,a}(z)| \leq \lambda(|\eta| + c) =: B_0$. Hence, the DAT discriminant equals a logit term plus a (scaled) log-likelihood term, up to a bounded remainder. With equal priors, argmax over $(y, a)$ is Bayes-aligned.*

**Corollary 3.4** (DAT reinstates anisotropy under Kent). *Under Assumption 3.2 and the Kent (Fisher-Bingham) model for the group density, the DAT margin admits the decomposition*

$$m(z) \approx \underbrace{\big(\tau\, w + \alpha\lambda\, \kappa\, \gamma_1\big)^\top z}_{\text{linear (axial) part}}$$
$$+ \underbrace{\alpha\lambda\, \beta\big[(\gamma_2^\top z)^2 - (\gamma_3^\top z)^2\big]}_{\text{anisotropy correction}} - \alpha\lambda \log c_d(\kappa, \beta).$$

*where the constant terms (including $r_{y,a}(z)$) do not affect the argmax. Thus, when $\beta \neq 0$, DAT adds the missing quadratic, anisotropy-sensitive term that pure cosine scoring ignores, correcting its bias in elliptical embeddings.*

Proofs of Proposition 3.1, Theorem 3.3, and Corollary 3.4 are detailed in Appendix A.

**Beyond Bayes alignment.** DAT also admits a surrogate-risk interpretation. For two groups $g = (y, a)$ and $g' = (y', a')$, DAT changes the pairwise margin as

$$m_g(z) - m_{g'}(z) = \alpha\lambda\Delta \log p(z) + \tau\Delta w^\top z + \Delta r(z),$$

*Table 1.* Zero-shot classification results on Waterbirds using the Worst-Group accuracy (WG), Average accuracy (Avg), and Gap between Average and Worst-Group accuracy (Gap). Higher WG (↑) and Avg, and lower Gap (↓) values are better. The best and second-best results are highlighted in **bold** and underlined, respectively.

| Method | CLIP (ViT-B/32) | | | CLIP (ViT-L/14) | | | CLIP (ResNet50) | | |
|---|---|---|---|---|---|---|---|---|---|
| | WG | Avg | Gap | WG | Avg | Gap | WG | Avg | Gap |
| ZS | 41.37 | 68.48 | 27.11 | 31.93 | 83.72 | 51.79 | 35.36 | 80.64 | 45.28 |
| Group Prompt | 43.46 | 66.79 | 23.33 | 10.44 | 56.12 | 45.68 | 49.84 | 70.96 | 21.12 |
| Ideal words | 60.28 | 79.20 | 18.92 | 64.17 | 87.67 | 23.50 | 39.90 | 79.48 | 40.39 |
| Orth-Cali | 54.99 | 69.19 | 14.20 | 58.86 | 86.31 | 27.45 | 60.84 | 84.47 | 19.67 |
| Perception CLIP | 59.78 | **82.50** | 22.72 | 54.12 | 86.74 | 32.62 | 48.21 | **91.51** | 43.30 |
| ROBOSHOT | 54.41 | 71.92 | 17.51 | 45.17 | 64.43 | 19.26 | 69.60 | 96.05 | 42.45 |
| TIE | 71.35 | 79.82 | 8.47 | 78.82 | 84.12 | **5.30** | 52.96 | 83.62 | 30.66 |
| TIE* | 61.24 | 76.91 | 15.67 | 61.60 | 78.98 | 17.38 | 34.11 | 81.19 | 47.08 |
| DAT | **75.08** | 80.36 | **5.28** | **83.33** | **89.57** | 6.42 | **75.08** | 83.83 | **8.75** |
| DAT* | 64.02 | 82.33 | 18.31 | 79.75 | 87.87 | 8.12 | 63.71 | 82.65 | 18.94 |

where $\Delta \log p(z) = \log p_g(z) - \log p_{g'}(z)$, $\Delta w = w_g - w_{g'}$, and $\Delta r(z) = r_g(z) - r_{g'}(z)$. Thus, when the density gap dominates the similarity and bounded-remainder terms, DAT can increase the margin toward the Bayes-preferred group under the density model and reduce logistic/hinge surrogate risks. Formal statements and proofs are provided in Appendix A.4.

## 4. Experiments

Following Adila et al. (2024); Lu et al. (2025), we conduct zero-shot classification experiments on benchmark datasets designed to test spurious correlations. We compare DAT against standard zero-shot classification and recent zero-shot debiasing baselines across multiple datasets and backbones.

**Datasets and Models.** We use Waterbirds (Sagawa et al., 2020), CelebA (Liu et al., 2015), COVID-19 (Cohen et al., 2020), and FMoW (Christie et al., 2018). Further details about the datasets are provided in Appendix B.4. Our evaluation covers multiple vision–language models: three CLIP variants (ViT-B/32, ViT-L/14, ResNet-50) (Radford et al., 2021), as well as ALIGN and AltCLIP. For COVID-19, we adopt BiomedCLIP (Zhang et al., 2023), a CLIP variant fine-tuned on biomedical data. Following Lu et al. (2025), we use CLIP ViT-L/14 for FMoW due to the dataset's complexity. Across all settings, experiments are performed using frozen embeddings from the pre-trained models.

**Metric and Baselines.** We use three metrics in our experiments: average accuracy % (Avg), Worst-Group accuracy % (WG), and the gap between the two % (Gap), which are evaluated in many spurious correlation works (Adila et al., 2024; Lu et al., 2025). A robust model should have a high Avg and WG, with a small Gap between them. In all result tables, the best score is highlighted in bold, and the second-best score

is underlined. We benchmark our approach against both simple baselines and the latest methods in robust zero-shot classification. Specifically, the baselines consist of standard zero-shot classification (ZS) and a variant that incorporates group information through prompting (Group prompt). For comparison with prior work, we include recent state-of-the-art approaches such as Ideal words (Trager et al., 2023), Orth-Cali (Chuang et al., 2023), Perception CLIP (An et al., 2024), ROBOSHOT (Adila et al., 2024), and TIE/TIE* (Lu et al., 2025), which are detailed in Section 2.

**Prompt Details for Reproducibility.** In zero-shot classification, we employ three categories of text prompts: label prompts, spurious prompts, and group prompts. For spurious prompts, we utilized those provided by Lu et al. (2025), and for group prompts, we concatenated the class and attribute templates. To support reproducibility, the full list of all types of prompts used in our experiments is included in Appendix B.5.[1]

**Settings.** For DAT, the neighbourhood size $k$ is set to 10 on Waterbirds, CelebA, and COVID-19, and to 30 on FMoW. The number of reference samples per group $n$ is 56 for Waterbirds, 128 for CelebA, 40 for COVID-19, and 50 for FMoW. The scaling parameter $\lambda$ is set to 10 for Waterbirds, COVID-19, and FMoW, and 1 for CelebA. For DAT*, we use the same setting as DAT, except on Waterbirds, where we set $\lambda = 1$. For Waterbirds, COVID-19, and FMoW, we construct the reference sets from the training split. For CelebA, we use the validation split, which provides enough group coverage. We utilized an NVIDIA H100 GPU with frozen weights. Details on implementation efficiency, in comparison to TIE, are provided in Appendix B.5, where we show that DAT achieves higher efficiency.

---

[1]Code available at https://github.com/AfsanehEB/DAT

*Table 2.* Zero-shot classification results on CelebA using the Worst-Group accuracy (WG), Average accuracy (Avg), and Gap between Average and Worst-Group accuracy (Gap). Higher WG ($\uparrow$) and Avg, and lower Gap ($\downarrow$) values are better. The best and second-best results are highlighted in **bold** and underlined, respectively.

| Method | CLIP (ViT-B/32) | | | CLIP (ViT-L/14) | | | CLIP (ResNet50) | | |
|---|---|---|---|---|---|---|---|---|---|
| | WG | Avg | Gap | WG | Avg | Gap | WG | Avg | Gap |
| ZS | 78.89 | 84.27 | 5.38 | 73.35 | 81.20 | 7.85 | 69.69 | 81.58 | 11.89 |
| Group Prompt | 74.90 | 80.38 | 5.48 | 68.94 | 77.86 | 8.92 | 70.59 | 79.48 | 8.89 |
| Ideal words | 78.12 | 80.96 | 2.84 | 76.67 | **89.15** | 12.48 | 65.65 | 76.27 | 10.62 |
| Orth-Cali | 77.92 | 82.31 | 4.39 | 77.69 | 81.39 | 3.70 | 69.13 | 76.47 | 7.34 |
| Perception CLIP | 76.46 | 80.32 | 3.86 | 78.70 | 81.41 | 2.71 | 80.22 | 85.17 | **4.95** |
| ROBOSHOT | 80.52 | 84.77 | 4.25 | 82.61 | 85.54 | 2.93 | 73.96 | 80.90 | 6.94 |
| TIE | **82.63** | 85.11 | **2.48** | 84.60 | 86.17 | 1.57 | 75.32 | 81.71 | 6.39 |
| TIE* | 82.61 | 85.10 | 2.49 | 81.98 | 84.27 | 2.29 | 75.30 | 81.70 | 6.40 |
| DAT | 78.53 | 87.09 | 8.56 | **84.94** | 86.54 | **1.19** | **80.79** | 87.09 | 6.30 |
| DAT* | 78.53 | **87.11** | 8.58 | 84.93 | 86.54 | 1.61 | 78.53 | **88.29** | 9.76 |

*Table 3.* Zero-shot classification results on COVID-19 (medical dataset) and FMoW (multi-label dataset).

*(a)* COVID-19 (BiomedCLIP).

| COVID-19 | | | |
|---|---|---|---|
| Method | WG | Avg | Gap |
| ZS | 44.83 | 61.81 | 16.98 |
| Group Prompt | 27.58 | 48.27 | 20.69 |
| Ideal words | 23.53 | 56.84 | 33.31 |
| Orth-Cali | 44.83 | 51.72 | 6.89 |
| Perception CLIP | 48.84 | 56.87 | 8.03 |
| ROBOSHOT | 32.75 | 53.10 | 20.35 |
| TIE | 52.17 | 62.50 | 10.33 |
| TIE* | 50.22 | 61.08 | 10.86 |
| DAT | 65.22 | **75.69** | 10.47 |
| DAT* | **72.41** | 74.30 | **1.89** |

*(b)* FMoW (ViT-L/14).

| FMoW | | | |
|---|---|---|---|
| Method | WG | Avg | Gap |
| ZS | 18.06 | 26.02 | 7.96 |
| Group Prompt | 8.75 | 14.69 | 5.94 |
| Ideal words | 11.14 | 20.21 | 9.07 |
| Orth-Cali | 19.45 | 26.11 | 6.66 |
| Perception CLIP | 12.61 | 17.70 | 5.09 |
| ROBOSHOT | 10.88 | 19.79 | 8.91 |
| TIE | 20.19 | 26.62 | 6.43 |
| TIE* | 19.84 | 26.65 | 6.81 |
| DAT | **27.75** | **31.19** | **3.44** |
| DAT* | 18.63 | 29.56 | 10.93 |

### 4.1. Results

**Waterbirds and CelebA.** As shown in Table 1, DAT achieves the highest WG accuracy across all baselines, surpassing the strongest prior by roughly 4-14%, depending on the backbone. The largest gains are observed with ViT-L/14 and ResNet-50 backbones. DAT* also achieves competitive performance, particularly with ViT-based models, and maintains comparable average accuracy. Table 2 reports zero-shot results on CelebA. Consistent with the trends observed in Table 1, DAT and DAT* achieve the strongest performance across most metrics, particularly in terms of WG accuracy. Moreover, DAT* attains results comparable to DAT, with only a minor difference.

**Medical domain and Multi-Label.** We further evaluate DAT on medical imaging using COVID-19 chest X-ray datasets with BiomedCLIP. As shown in Table 3a, DAT and DAT* outperform all baselines, achieving over 20% higher WG than the latest baselines and reducing Gap by 5%. To assess scalability to richer label and attribute structure, we

also test on FMoW, which has 62 classes, and 5 spurious attributes. Table 3b shows that DAT attains the highest WG and Avg, and markedly reduces the Gap. The label-free variant, DAT*, is competitive as well, especially on WG and Avg.

**Evaluation of Other Models.** To evaluate the effectiveness of our method across a broader range of VLMs, we also test DAT/DAT* on ALIGN and AltCLIP. For a fair comparison, baseline results are taken from Adila et al. (2024), which is one of the latest baselines that evaluates these two models using these datasets. As shown in Table 4, on both Waterbirds and CelebA, DAT and DAT* in most cases outperform previous methods, with a remarkable improvement in WG accuracy, highlighting robustness across distinct VLMs.

### 4.2. Ablations

**Effect of Parameters.** We investigate the sensitivity of DAT to the scaling factor ($\lambda$), the number of samples per group ($n$), and the number of neighbours for density estimation ($k$).

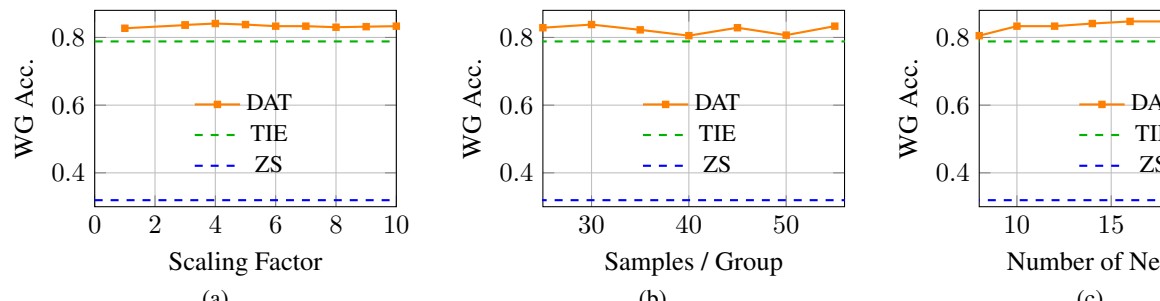

*Figure 3.* Comparison of WG accuracy under varying (a) scaling factor, (b) number of samples per group, and (c) number of neighbours using Waterbirds and CLIP(ViT-L/14).

*Table 4.* Generalisation of DAT to other models: zero-shot classification results using ALIGN and AltCLIP on Waterbirds and CelebA datasets. The best and second-best results are highlighted in **bold** and underlined, respectively.

| | Waterbirds | | | | | | CelebA | | | | | |
|---|---|---|---|---|---|---|---|---|---|---|---|---|
| | ALIGN | | | AltCLIP | | | ALIGN | | | AltCLIP | | |
| Method | WG | Avg | Gap | WG | Avg | Gap | WG | Avg | Gap | WG | Avg | Gap |
| ZS | 50.3 | 72.0 | 21.7 | 35.8 | 90.1 | 54.3 | 77.2 | 81.8 | 4.6 | 79.7 | 82.3 | **2.6** |
| Group Prompt | 5.8 | 72.5 | 66.7 | 29.4 | 82.4 | 53.0 | 67.4 | 78.3 | 10.9 | 79.0 | 82.3 | 3.3 |
| ROBOSHOT | 41.0 | 50.9 | 9.9 | 54.8 | 78.5 | 23.7 | **83.4** | **86.3** | 2.9 | 77.2 | 86.0 | 8.8 |
| DAT | **73.36** | **82.90** | **9.54** | **81.31** | **91.54** | 10.23 | 82.76 | 84.82 | **2.06** | 83.89 | 86.64 | 2.75 |
| DAT* | 63.72 | 76.16 | 12.44 | 56.23 | 89.76 | 33.53 | 82.69 | 84.80 | 2.11 | 83.89 | **86.68** | 2.79 |

Results in Figure 3 show that DAT remains stable across a wide range of settings, consistently outperforming the strongest baseline (TIE), as well as the simplest baseline (zero-shot classification).

**Prompt Template and Object Description.** In Appendix E.1–E.2, we show that DAT* is largely insensitive to prompt template phrasing, while the semantic granularity of object terms plays a more important role, with fine-grained descriptions consistently improving worst-group performance.

**Using Different Terms of Aggregation.** We further ablate the two scoring terms used in DAT. The group-prompt score corresponds to the group-specific corrected score $\tilde{s}_{y,a}$. As shown in Figure 4, using either $\tilde{s}_{y,a}$ or $\tilde{s}_{y,\text{Avg}}$ alone improves over the baselines, but neither consistently matches the full DAT formulation. The group-specific score $\tilde{s}_{y,a}$ directly corrects subgroup-level density imbalance, but can be less stable because prediction relies only on subgroup-specific corrections. In contrast, the class-marginal score $\tilde{s}_{y,\text{Avg}}$ provides a more balanced aggregation across attributes, but does not fully exploit subgroup-specific corrections. Combining both terms yields the strongest overall worst-group performance, indicating that the two components provide complementary benefits.

**Robust Text Prompt.** Following An et al. (2024); Lu et al. (2025), we expand group descriptions by incorporating multiple semantically related variants of spurious attributes (e.g., alternative ways of describing land or water backgrounds in

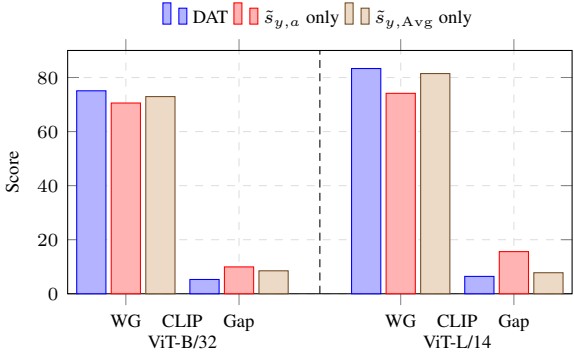

*Figure 4.* Aggregation ablation on Waterbirds. The full DAT aggregation outperforms using either corrected score alone.

Waterbirds, as generated in Liang et al. (2022)) and averaging their embeddings. Spurious attribute variants and corresponding group prompts are provided in Tables 11 and 12 in Appendix D.1. This robustified representation is intended to better capture attribute variability and has been shown to improve both WG and Avg accuracy across backbones. As shown in Table 5, however, group-robust prompts result in only marginal changes, minor improvements in some cases, but overall negligible differences. In contrast, the performance differences between DAT* and DAT* Robust are more pronounced. This aligns with findings from Lu et al. (2025), since DAT* uses the spurious prompts directly to construct group references. Consequently, varying the phrasing of spurious attributes has a greater impact, often

*Table 5.* Group robustify prompting evaluation using the Waterbirds dataset.

| Method | ViT-B/32 | | | ViT-L/14 | | | ResNet-50 | | |
|---|---|---|---|---|---|---|---|---|---|
| | WG | Avg | Gap | WG | Avg | Gap | WG | Avg | Gap |
| DAT | **75.08** | **80.36** | 5.28 | 83.33 | **89.57** | 6.42 | 75.08 | **83.83** | 8.75 |
| DAT Robust | 74.99 | 80.12 | **5.13** | **83.49** | 89.09 | **5.60** | **75.39** | 83.65 | **8.26** |
| DAT* | 64.02 | **82.33** | 18.31 | **79.75** | **87.87** | 8.12 | 63.71 | **82.65** | 18.94 |
| DAT* Robust | **74.68** | 78.82 | **4.14** | 79.64 | 84.62 | **4.98** | **69.00** | 73.85 | **4.85** |

improving WG and reducing the GAP, indicating that using multiple semantically aligned prompt variants generally leads to enhanced or comparable performance. A similar trend is observed on the CelebA dataset; corresponding results are reported in Appendix D.2.

Additional analyses on reference-set quality and global normalisation baselines are provided in Appendices C.3 and C.2.

## 5. Conclusion

We introduced the Density-Aware Translation method, a simple zero-shot mechanism that rescales image-text similarities using a geometric density proxy computed from small group references, addressing the bias introduced by the ellipsoidal shape of CLIP embeddings. Across multiple datasets, models, and settings, we demonstrated that DAT consistently improves performance, raising WG and Avg accuracy while reducing the gap between them. From a theoretical perspective, we showed that DAT's multiplicative correction leads to an additive discriminant that aligns cosine similarity with a Bayes-style log-likelihood, while reinstating anisotropy-sensitive terms that cosine similarity alone overlooks. Although DAT and its variant DAT* perform strongly compared to prior methods, we believe future work should explore adaptive multimodal approaches for density estimation in the embedding space that are less sensitive to reference selection and prompt design.

## Impact Statement

This work aims to mitigate spurious correlations in vision–language models to enhance robustness across diverse groups. By reducing sensitivity to spurious correlations, the proposed method aims to improve model performance across underrepresented groups without requiring additional supervision or fine-tuning. The broader societal implications of this work are aligned with those commonly associated with improved robustness and reliability in machine learning systems, and we do not identify any additional ethical concerns that require specific discussion.

## Acknowledgements

This research was supported by The University of Melbourne's Research Computing Services, the Petascale Campus Initiative, and the Spartan HPC facilities. This Facility was established with the assistance of LIEF Grant LE170100200. Moreover, this research was supported by the ARC Centre of Excellence for Automated Decision-Making and Society (CE200100005), and funded partially by the Australian Government through the Australian Research Council.

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

# Appendix

# A. Theoretical Analysis and Proofs

### A.1. Proof of proposition 3.1

*Proof.* Fix any $t \in (-1, 1)$ and $r \in \left(0, \sqrt{1 - t^2}\right)$. Define

$$\gamma_1^\top z_\pm = t, \qquad \gamma_2^\top z_+ = r, \ \gamma_3^\top z_+ = 0, \qquad \gamma_2^\top z_- = 0, \ \gamma_3^\top z_- = r,$$

and take remaining coordinates 0 so that $z_\pm \in \mathbb{S}^{d-1}$. Then $w^\top z_+ = w^\top z_- = t$ (same cosine), while

$$\log p(z_+) - \log p(z_-) = \beta \left[ (\gamma_2^\top z_+)^2 - (\gamma_3^\top z_+)^2 - \left( (\gamma_2^\top z_-)^2 - (\gamma_3^\top z_-)^2 \right) \right] = 2\beta r^2 \neq 0.$$

Thus $\log p$ strictly prefers one of $z_\pm$ (sign set by $\beta$), whereas cosine ties them. $\square$

### A.2. Proof of Theorem 3.3

*Proof.* By the definition of the DAT margin,

$$m_{y,a}(z) \ = \ \tau \, w_{y,a}^\top z \ - \ \lambda \log D_{y,a}(z).$$

Based on Assumption 3.2, which states that for some $\alpha > 0$, $\eta \in \mathbb{R}$ and a bounded error $\epsilon_{y,a}(z)$ with $|\epsilon_{y,a}(z)| \leq c$,

$$\log D_{y,a}(z) \ = \ \alpha\big( -\log p_{y,a}(z) \big) \ + \ \eta \ + \ \epsilon_{y,a}(z).$$

Substituting this into the margin gives

$$m_{y,a}(z) = \tau \, w_{y,a}^\top z \ - \ \lambda \Big\{ \alpha\big( -\log p_{y,a}(z) \big) + \eta + \epsilon_{y,a}(z) \Big\}$$
$$= \tau \, w_{y,a}^\top z \ + \ \alpha\lambda \log p_{y,a}(z) \ - \ \lambda\eta \ - \ \lambda \, \epsilon_{y,a}(z).$$

Define the remainder

$$r_{y,a}(z) \ := \ -\lambda\eta - \lambda \, \epsilon_{y,a}(z),$$

so that

$$m_{y,a}(z) \ = \ \tau \, w_{y,a}^\top z \ + \ \alpha\lambda \log p_{y,a}(z) \ + \ r_{y,a}(z).$$

Because $|\epsilon_{y,a}(z)| \leq c$, we have the uniform bound

$$|r_{y,a}(z)| \ \leq \ \lambda\big( |\eta| + c \big) \ =: \ B_0.$$

This proves the stated decomposition and bound.

**Bayes alignment.** With equal priors, the Bayes rule ranks groups by $\log p_{y,a}(z)$. For any two groups $g = (y, a)$ and $g' = (y', a')$,

$$m_g(z) - m_{g'}(z) \ = \ \alpha\lambda\Big( \log p_g(z) - \log p_{g'}(z) \Big) \ + \ \tau\big( w_g - w_{g'} \big)^\top z \ + \ r_g(z) - r_{g'}(z).$$

Hence $m$ ranks identically to the Bayes score whenever the Bayes gap dominates the bounded perturbations, e.g.

$$\alpha\lambda \,\big| \log p_g(z) - \log p_{g'}(z) \big| \ > \ \big| \tau \, (w_g - w_{g'})^\top z \big| \ + \ |r_g(z)| + |r_{g'}(z)|,$$

and in particular when $\tau = 0$ (or the $\tau$-term is treated as a fixed bias) and $B_0$ is small. Thus, the DAT discriminant equals a similarity term plus a (scaled) log-likelihood term up to a bounded remainder, and is Bayes-aligned in the argmax under equal priors in the sense above. $\square$

### A.3. Proof of Corollary 3.4

*Proof sketch.* By Theorem 3.3, $m(z) \ = \ \tau \, w^\top z + \alpha\lambda \log p(z) + r(z)$. Substitute the Kent log-density; collect linear ($\propto \gamma_1^\top z$), quadratic ($\propto (\gamma_2^\top z)^2 - (\gamma_3^\top z)^2$), and constant terms. Absorb bounded $r(z)$ and constants into a bias; these do not change the argmax. $\square$

### A.4. Groupwise Surrogate-Risk Improvement

For a group $g$ (e.g., a specific $(y, a)$), let $Z \in \mathbb{S}^{d-1}$ denote its image embedding and $Y \in \{\pm 1\}$ the group label in a one-vs-rest reduction.[2] Recall from Theorem 3.3 (main text) the DAT margin decomposition

$$m(Z) = m_0(Z) + \alpha\lambda\, L(Z) + r(Z), \qquad m_0(Z) := \tau\, w_g^\top Z, \quad L(Z) := \log p_g(Z),$$

where $\alpha\lambda > 0$ is the density weight and $r$ is a bounded remainder.

For a convex surrogate $\ell$, the groupwise surrogate risk is

$$R_\ell^{(g)}(m) := \mathbb{E}\big[\ell\big(Y\, m(Z)\big) \,\big|\, G = g\big], \qquad \ell \in \{\ell_{\log}, \ell_{\text{hinge}}\},$$

with $\ell_{\log}(t) = \log(1 + e^{-t})$ and $\ell_{\text{hinge}}(t) = \max\{0, 1 - t\}$.

**Assumption A.1** (Hard-mass nondegeneracy). *There exists $c_g > 0$ such that for the path*

$$m_\theta(Z) := m_0(Z) + \theta\big(\alpha\lambda\, L(Z) + r(Z)\big), \qquad \theta \in [0, 1],$$

*we have, for all $\theta \in [0, 1]$,*

$$\mathbb{E}\big[\sigma\big(-Y\, m_\theta(Z)\big)\, Y\, L(Z) \,\big|\, G = g\big] \geq c_g \quad \text{and} \quad \mathbb{E}\big[\mathbf{1}\{Y\, m_\theta(Z) < 1\}\, Y\, L(Z) \,\big|\, G = g\big] \geq c_g,$$

*where $\sigma(t) = 1/(1 + e^{-t})$ is the logistic sigmoid, and there exists $B_g < \infty$ such that $|r(Z)| \leq B_g$ almost surely.*

**Theorem A.2** (Strict groupwise surrogate-risk decrease). *Under Assumptions A.1, if $\alpha\lambda\, c_g > B_g$, then for $\ell \in \{logistic, hinge\}$,*

$$R_\ell^{(g)}(m) < R_\ell^{(g)}(m_0).$$

*Proof.* **Logistic.** We have:

$$\frac{d}{d\theta} R_{\log}^{(g)}(m_\theta) = \mathbb{E}\big[\ell_{\log}'(Y m_\theta(Z))\, Y\big(\alpha\lambda L(Z) + r(Z)\big) \,\big|\, G = g\big] = -\mathbb{E}\big[\sigma\big(-Y m_\theta(Z)\big)\, Y\big(\alpha\lambda L(Z) + r(Z)\big)\big].$$

$$\frac{d}{d\theta} R_{\log}^{(g)}(m_\theta) = -\alpha\lambda\, \mathbb{E}[\sigma(-Y m_\theta)\, Y\, L] - \mathbb{E}[\sigma(-Y m_\theta)\, Y\, r].$$

Assumption A.1 gives the first expectation $\geq c_g$, while $|\mathbb{E}[\sigma(-Y m_\theta)\, Y\, r]| \leq \mathbb{E}[|r|] \leq B_g$ a.s. Hence $\frac{d}{d\theta} R_{\log}^{(g)}(m_\theta) \leq -\alpha\lambda c_g + B_g < 0$.

**Hinge.** For the hinge loss:

$$\frac{d}{d\theta} R_{\text{hinge}}^{(g)}(m_\theta) \in -\mathbb{E}\big[\mathbf{1}\{Y m_\theta(Z) < 1\}\, Y\big(\alpha\lambda L(Z) + r(Z)\big) \,\big|\, G = g\big],$$

where the right-hand side is any measurable subderivative (a.e. well-defined).

$$\frac{d}{d\theta} R_{\text{hinge}}^{(g)}(m_\theta) = -\alpha\lambda\, \mathbb{E}[\mathbf{1}\{Y m_\theta < 1\}\, Y\, L] - \mathbb{E}[\mathbf{1}\{Y m_\theta < 1\}\, Y\, r].$$

Assumption A.1 lower-bounds the first term by $c_g$, while $|\mathbb{E}[\cdot]| \leq B_g$ for the second. Thus $\frac{d}{d\theta} R_{\text{hinge}}^{(g)}(m_\theta) \leq -\alpha\lambda c_g + B_g < 0$ and integration finishes the proof. $\qquad\square$

## B. Algorithm and Experimental Details

### B.1. Algorithm

The algorithms for DAT and DAT* are presented in Algorithms 1 and 2, respectively. Their primary distinction lies in the construction of the reference sets: DAT assumes direct access to the spurious attribute, whereas DAT* infers it through zero-shot classification.

---

[2]The multiclass (multi-group) case follows by standard one-vs-rest aggregation; we state the binary reduction for clarity.

---

**Algorithm 1** DAT

---

**Input:** Image $x$, Image encoder $\phi_I(\cdot)$, Text encoder $\phi_T(\cdot)$, Group prompts $\{t_{y,a}\}$, Class-only prompts $\{t_y\}$, Reference sets $\{R_{y,a}\}$.

**Output:** Predicted label $\hat{y}$.

1: $z \leftarrow \phi_I(x)$ {Compute image embedding}
2: **for** each group $(y, a)$ **do**
3:    $s_{y,a}(x) \leftarrow \langle z, \phi_T(t_{y,a}) \rangle$ {Compute group-wise similarity}
4:    **if** $|R_{y,a}| < n$ **then**
5:       $D_{y,a}(z) \leftarrow \infty$ {Insufficient reference samples}
6:    **else**
7:       $D_{y,a}(z) \leftarrow \text{SLOF}(z; R_{y,a})$ {Local density}
8:    **end if**
9:    $\tilde{s}_{y,a}(x) \leftarrow \frac{s_{y,a}(x)}{(D_{y,a}(z)+\varepsilon)^\lambda}$ {DAT correction}
10: **end for**
11: $\tilde{s}_{y,\text{Avg}}(x) \leftarrow \frac{1}{M+1}\Big( \sum_{a \in \mathcal{A}} \tilde{s}_{y,a}(x) + s_y(x) \Big)$ {Aggregate class-marginal score}
12: $\hat{y} \leftarrow \arg\max_{y \in \mathcal{Y}} \max_{a \in \mathcal{A}} \{\tilde{s}_{y,a}(x), \tilde{s}_{y,\text{Avg}}(x)\}$
13: **return** $\hat{y}$

---

---

**Algorithm 2** DAT*

---

**Input:** Image $x$, Image encoder $\phi_I(\cdot)$, Text encoder $\phi_T(\cdot)$, Group prompts $\{t_{y,a}\}$, Class-only prompts $\{t_y\}$, Spurious prompts $\{t_y\}$, Reference sets $\{R_{y,a}\}$.

**Output:** Predicted label $\hat{y}$.

1: $z \leftarrow \phi_I(x)$ {Compute image embedding}
2: $\hat{a} = \arg\max_{a \in \mathcal{A}} \langle z, w_a \rangle$. {Infer spurious attributes}
3: Make the reference sets $\{R_{y,a}\}$
4: **for** each group $(y, a)$ **do**
5:    $s_{y,a}(x) \leftarrow \langle z, \phi_T(t_{y,a}) \rangle$ {Compute group-wise similarity}
6:    **if** $|R_{y,a}| < n$ **then**
7:       $D_{y,a}(z) \leftarrow \infty$ {Insufficient reference samples}
8:    **else**
9:       $D_{y,a}(z) \leftarrow \text{SLOF}(z; R_{y,a})$ {Local density}
10:   **end if**
11:   $\tilde{s}_{y,a}(x) \leftarrow \frac{s_{y,a}(x)}{(D_{y,a}(z)+\varepsilon)^\lambda}$ {DAT correction}
12: **end for**
13: $\tilde{s}_{y,\text{Avg}}(x) \leftarrow \frac{1}{M+1}\Big( \sum_{a \in \mathcal{A}} \tilde{s}_{y,a}(x) + s_y(x) \Big)$ {Aggregate class-marginal score}
14: $\hat{y} \leftarrow \arg\max_{y \in \mathcal{Y}} \max_{a \in \mathcal{A}} \{\tilde{s}_{y,a}(x), \tilde{s}_{y,\text{Avg}}(x)\}$
15: **return** $\hat{y}$

---

### B.2. Reference-set construction

Given a group pool of images $X_{y,a} = \{x_{y,a}^{(h)}\}_{h=1}^{N_{y,a}}$, let the (unit–norm) image embeddings be $z_{y,a}^{(h)} = \phi_I(x_{y,a}^{(h)})/\|\phi_I(x_{y,a}^{(h)})\|_2$ and define $\mathcal{G}_{y,a}^{feat} = \{z_{y,a}^{(h)}\}_{h=1}^{N_{y,a}} \subset \mathbb{R}^d$. The group mean in feature space is

$$\mu_{y,a} \;=\; \frac{1}{N_{y,a}} \sum_{h=1}^{N_{y,a}} z_{y,a}^{(h)}.$$

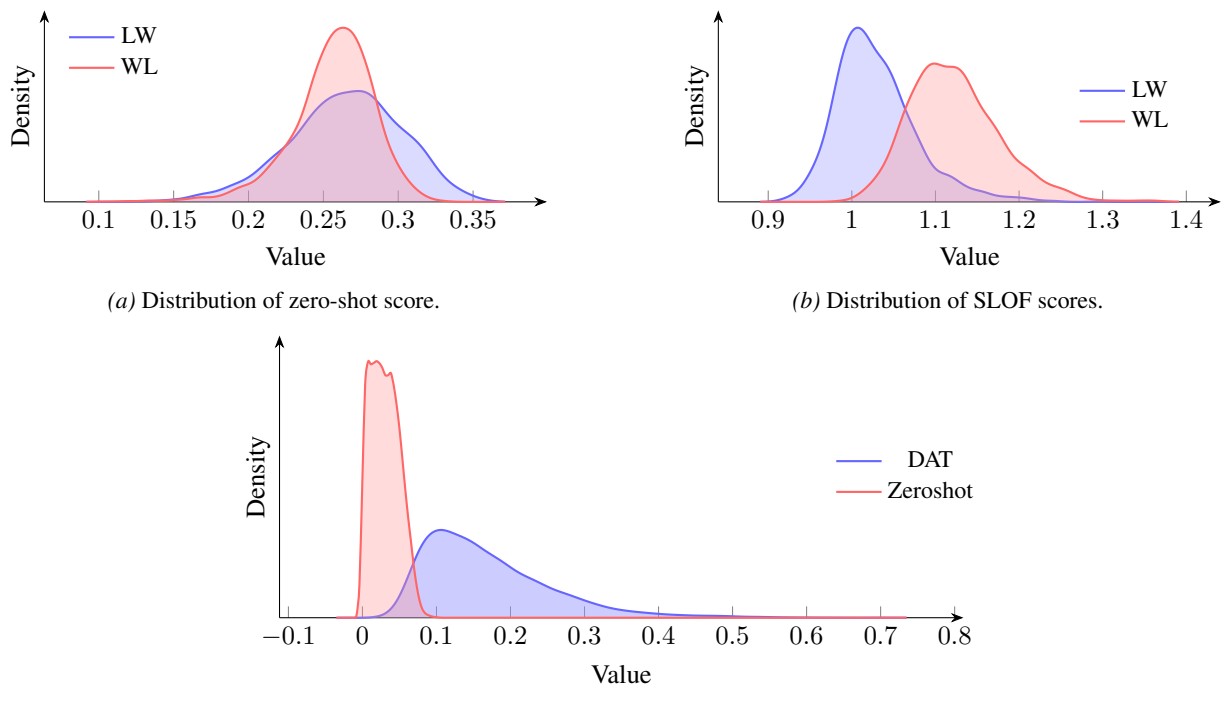

*(a)* Distribution of zero-shot score.

*(b)* Distribution of SLOF scores.

*(c)* Score difference distributions.

*Figure 5.* Density comparisons on Waterbirds with CLIP ViT-L/14. (a) For WL images (waterbird on land), raw cosine similarity to the WL vs LW prompts shows heavy overlap, and cosine alone cannot reliably separate the true group from its spuriously aligned counterpart. (b) SLOF values for the same WL images, measured against WL vs LW references; WL assigns a higher SLOF (sparser) than LW, revealing density asymmetry. (c) Across the dataset, DAT enlarges the max–min group score gap relative to the baseline, yielding more decisive predictions.

Starting from an empty reference set $R_{y,a}^{(0)} = \varnothing$ with running sum $S_0 = \mathbf{0} \in \mathbb{R}^d$, we select a uniform budget of $n$ exemplars by greedy feature-space herding:

$$p_k^{(y,a)} \in \arg \min_{z \in \mathcal{G}_{y,a} \setminus R_{y,a}^{(k-1)}} \left\| \mu_{y,a} - \frac{1}{k} \left( z + \sum_{j=1}^{k-1} p_j^{(y,a)} \right) \right\|_2, \qquad k = 1, \ldots, n,$$

where we set $R_{y,a}^{(k)} = R_{y,a}^{(k-1)} \cup \{p_k^{(y,a)}\}$ and $S_{k-1} = \sum_{j=1}^{k-1} p_j^{(y,a)}$. Equivalently, the selection can be written as

$$p_k^{(y,a)} \in \arg \max_{z \in \mathcal{G}_{y,a}^{feat} \setminus R_{y,a}^{(k-1)}} \left( \langle z_{y,a}, \mu_{y,a} \rangle - \frac{1}{k} \langle z_{y,a}, S_{k-1} \rangle \right).$$

We break ties deterministically (by smallest index) for reproducibility.

*Table 6.* Prompts details used for different datasets.

| Dataset | Label prompts | Spurious prompts | Group prompts |
|---|---|---|---|
| Waterbirds | a photo of a landbird, a photo of a waterbird | a photo with a water background, a photo with a land background | a photo of a landbird in water, a photo of a waterbird in land, a photo of a landbird in land, a photo of a waterbird in water |
| CelebA | a photo of a celebrity with dark hair, a photo of a celebrity with blonde hair | a photo of a female, a photo of a male | a photo of a male celebrity with dark hair, a photo of a female celebrity with blonde hair, a photo of a Female celebrity with dark hair, a photo of a Male celebrity with blonde hair |
| COVID-19 | An X-ray image of a chest without Pneumonia, An X-ray image of a chest with Pneumonia | An X-ray image from a female, An X-ray image from a male | an X-ray image of a chest without Pneumonia from a male, an X-ray image of a chest with Pneumonia from a female, an X-ray image of a chest without Pneumonia from a female, an X-ray image of a chest with Pneumonia from a male |
| FMoW | A satellite image of a/an $y_{i_{i=0}^{i=61}}$ | Over Europe, Over Asia, Over Americas, Over Africa, Over Oceania | A satellite image of a/an $y_{i_{i=0}^{i=61}}$ over Europe, [A satellite image of a/an $y_{i_{i=0}^{i=61}}$ over Asia], [A satellite image of a/an $y_{i_{i=0}^{i=61}}$ over Americas], [A satellite image of a/an $y_{i_{i=0}^{i=61}}$ over Africa, [A satellite image of a/an $y_{i_{i=0}^{i=61}}$ over Oceania] |

### B.3. Zero-shot in Related Papers

In DAT, samples from the target domain are never observed during training, which is fully consistent with the standard definition of zero-shot learning in the literature (Wang et al., 2019; Lampert et al., 2013; Palatucci et al., 2009). The DAT* variant further allows construction of the reference set by inferring spurious features, without requiring target-domain supervision.

Our approach aligns with prior zero-shot spurious-correlation mitigation methods (Lu et al., 2025; An et al., 2024), while differing in its assumptions and reference-set construction.

In particular, **TIE** (Lu et al., 2025) computes translation scales from image embeddings over the target-domain training split, as reflected in its public implementation.

**PerceptionCLIP** (An et al., 2024) depends on contextual attribute construction; its semi-automated attribute-construction procedure can use large-scale image–text retrieval and the in-context reasoning capabilities of large language models.

**Zero-Shot Robustification** (Adila et al., 2024), in its label-free adaptation variant, assumes access to an unlabeled training set and a small labeled validation set to learn an additional embedding-space projection.

DAT uses only small, fixed reference exemplar sets constructed once via deterministic feature-space herding, and does not require large external data sources, global contextual retrieval, or embeddings from the full training set. As a result, the assumptions underlying DAT and DAT* are fully compatible with the zero-shot evaluation protocol adopted in prior work, and comparisons with TIE, PerceptionCLIP, and related baselines remain fair.

### B.4. Datasets

Our method, along with all baselines, is evaluated in the data sets listed below.

- **Waterbirds** (Koh et al., 2021; Sagawa et al., 2020): A binary bird classification task with labels $y \in \{\text{Landbird}, \text{Waterbird}\}$. The spurious attribute is the background type $a \in \{\text{Land}, \text{Water}\}$, where most landbirds appear on land and most waterbirds over water. This produces four groups: Landbird-Land, Landbird-Water, Waterbird-Land, and Waterbird-Water.

- **CelebA** (Liu et al., 2015): A large-scale face dataset (200K+ images) used for binary hair color classification $y \in \{\text{Dark hair}, \text{Blonde hair}\}$. Gender $a \in \{\text{Female}, \text{Male}\}$ is spuriously correlated with hair color, $94\%$ of blond-labeled images are female. Groups are: Female–Dark, Female–Blonde, Male–Dark, and Male–Blonde.

- **COVID-19** (Cohen et al., 2020): An X-ray dataset for pneumonia diagnosis, with task $y \in \{\text{No pneumonia}, \text{Pneumonia}\}$. Gender $a \in \{\text{Male}, \text{Female}\}$ serves as a spurious confounder. Groups include: Male–Pneumonia, Male–No pneumonia, Female–Pneumonia, and Female–No pneumonia.

- **FMoW** (Christie et al., 2018; Izmailov et al., 2021): A large-scale satellite dataset with 62 land-use/building classes. The spurious attribute is the geographical region $a \in \{\text{Africa}, \text{Americas}, \text{Asia}, \text{Europe}, \text{Oceania}\}$. Groups are defined purely by region. FMoW also exhibits a temporal domain shift; training images are collected before 2016, while validation and test images are collected in 2016-2017.

### B.5. Implementation and Reproducibility

**Details of Prompts.** For prompts, we followed (Liang et al., 2022) and utilized their proposed templates. We only created group prompts by simply combining two prompts, without adding any extra information. The prompt details are provided in Table 6.

**Implementation Efficiency.** We utilized an NVIDIA H100 GPU with frozen weights across all datasets. Table 7 reports the efficiency on the CelebA dataset, one of the large-scale datasets evaluated in this study. As shown, DAT demonstrates higher efficiency than TIE.

| Method | CLIP (ViT-B/32) | CLIP (ViT-L/14) | CLIP (ResNet50) |
|---|---|---|---|
| TIE | 970 | 947 | 951 |
| DAT | 203 | 148 | 153 |

*Table 7.* Comparison of computation time (seconds) efficiency using the CelebA dataset.

### B.6. Effect of Density Method

In this section, we study the impact of different density estimation methods within DAT, including SLOF, DAO (Dimensionality-Aware Outlier Detection), and $k$-NN distance. SLOF is used as the default density proxy in DAT and is described in detail in Section 3.3. Here, we provide a comparative analysis to justify this choice.

**DAO.** DAO (Anderberg et al., 2024) extends density-based outlier detection by explicitly incorporating local intrinsic dimensionality into the outlier scoring process. Building upon SLOF, DAO adjusts local density comparisons to account for variations in the underlying data geometry.

Formally, the DAO score for a query point $q$ is defined as

$$\text{DAO}_k(q) := \frac{1}{k} \sum_{o \in \text{NN}_k(q)} \left( \frac{k\text{-dist}(q)}{k\text{-dist}(o)} \right)^{\text{LID}^*_{F_o}},$$

where $k$-dist$(\cdot)$ denotes the distance to the $k$-th nearest neighbor, and $\text{LID}^*_{F_o}$ is the estimated local intrinsic dimensionality associated with neighbor $o$. The LID can be estimated using either the method of moments (MoM) (Amsaleg et al., 2018) or maximum likelihood estimation (MLE) (Levina & Bickel, 2004). A DAO score greater than one indicates that the query point is likely to be an outlier.

We compare SLOF, DAO (with both MLE and MoM LID estimation), and $k$-NN distance within the DAT framework on Waterbirds using CLIP ViT-B/32. Results are reported in Table 8.

DAT aims to correct samples that receive spuriously high zero-shot similarity scores due to alignment with spurious attributes. Therefore, the density proxy should answer the question: *Is this sample atypical relative to the group reference set, despite exhibiting high similarity to the group prompt?*

SLOF directly measures local sparsity relative to the reference group, effectively capturing whether a sample lies far from the core of the group distribution. In contrast, DAO assesses whether the observed deviation is large relative to the estimated local intrinsic dimensionality, which implicitly assumes that low-dimensional structure corresponds to normality.

As shown in Table 8, SLOF consistently outperforms DAO (under both MLE and MoM estimation) and $k$-NN distance across all values of $k$. This behavior is expected in our setting. In spurious correlation benchmarks, many spuriously aligned samples cluster tightly around the group mean, yielding low intrinsic dimensionality despite being semantically misleading. As a result, DAO tends to under-penalize such samples, whereas SLOF correctly assigns high sparsity scores due to their uneven distance distribution within the group.

Similarly, $k$-NN distance alone fails to distinguish between genuinely rare but semantically correct samples and spuriously frequent ones, as it lacks a relative normalization against neighbor densities. Overall, these results support SLOF as a more appropriate density proxy for DAT, as it directly targets non-uniformity in local neighborhood structure—the key signal required to suppress spurious alignment while preserving semantic correctness.

## C. Density Effect Visualization.

To better illustrate the role of SLOF in mitigating spurious correlations, we provide three complementary visualizations on the Waterbirds dataset with CLIP ViT-L/14.

In Figure 5a, we focus on the marginalized group WL (waterbird on land), which is often misclassified as LW(landbird on water). For these WL images, we plot the cosine similarity with respect to both group prompts *"a photo of a waterbird in land"* and *"a photo of a landbird in water"*. The strong overlap in distributions shows that raw cosine similarity based on group prompts alone cannot reliably separate the true group from its spuriously correlated counterpart.

*Table 8.* Performance comparison of different density methods on Waterbirds and CLIP (ViT-B/32).

| Method | WG | Avg | Gap |
|---|---|---|---|
| SLOF | **75.08** | 80.36 | **5.28** |
| DAO (MLE) ($k$=40) | 69.27 | 77.82 | 8.55 |
| DAO (MLE) ($k$=30) | 71.88 | 80.17 | 8.29 |
| DAO (MLE) ($k$=20) | 69.49 | 79.06 | 9.57 |
| DAO (MLE) ($k$=10) | 71.44 | 79.23 | 7.79 |
| DAO (MoM) ($k$=40) | 69.27 | 77.75 | 8.48 |
| DAO (MoM) ($k$=30) | 70.64 | 79.62 | 8.98 |
| DAO (MoM) ($k$=20) | 68.07 | 78.37 | 10.30 |
| DAO (MoM) ($k$=10) | 68.33 | 78.09 | 9.76 |
| $k$-NN ($k$=40) | 61.06 | 78.67 | 17.61 |
| $k$-NN ($k$=30) | 70.64 | 80.31 | 9.67 |
| $k$-NN ($k$=20) | 70.20 | **80.55** | 10.33 |
| $k$-NN ($k$=10) | 71.49 | 81.01 | 9.52 |

Figure 5b shows the SLOF values for the same WL images, this time measured relative to reference sets from WL and LW. Here, the WL references consistently assign higher SLOF values compared to LW, confirming that SLOF captures the sparsity structure of group embeddings and highlights the marginalization of rare groups.

Finally, Figure 5c visualizes the distribution of score differences (maximum minus minimum group score) across the dataset, comparing DAT with the uncorrected baseline. DAT widens this gap, producing more confident and reliable predictions.

Together, these results demonstrate that SLOF not only exposes the density imbalance between rare and spurious groups but also provides a mechanism for DAT to improve the discriminative margin in practice.

### C.1. Illustrating the Score and SLOF Distribution across Spurious Correlated Groups

We analyze how DAT influences prediction scores across groups in both the Waterbirds and CelebA datasets. As shown in Figure 6, the raw cosine similarity scores for the true group (Landbird in Water, LW) and the spuriously aligned group (Waterbird in Water, WW) are closely overlapping. This score overlap explains why LW images can be misclassified as WW. However, examining the SLOF distributions reveals that LW samples are consistently denser under their own group reference set and sparser under WW references. Despite the narrow SLOF range (due to low intra-group variance in Waterbirds), this density asymmetry is sufficient for DAT to recalibrate scores effectively.

A similar pattern appears in CelebA, as illustrated in Figure 7. The raw scores for the worst group (Dark-Hair Male, DH-M) and its spuriously correlated counterpart (Blond-Hair Male, BH-M) show substantial overlap, reflecting the dataset's strong hair–gender correlation. In contrast, SLOF scores offer clearer separation, with DH-M images exhibiting lower SLOF values under their correct reference group. The overall SLOF range is wider for CelebA due to higher embedding variance, but DAT's use of relative SLOF values within each group ensures stable and consistent behavior.

### C.2. Comparison with Global Normalisation

We further compare DAT with two simple global normalisation baselines, mean-centering (MC) and whitening (W). These methods reduce global distributional differences between image and text embeddings when sufficient samples from both modalities are available. However, DAT addresses a different source of bias by modelling local density differences between subgroups. The misalignment shown in Figure 1 arises from subgroup geometry rather than only global modality misalignment. Therefore, global normalisation alone may not correct the relative structure that causes spurious alignment.

Table 9 reports the results on Waterbirds. Mean-centering provides moderate improvements over zero-shot and group-prompt baselines, indicating that some global bias exists. Whitening does not consistently improve performance and in some cases, reduces average accuracy, likely because it removes global anisotropic structure that may also contain useful semantic information. In contrast, DAT achieves the highest worst-group accuracy across all backbones. These results show that

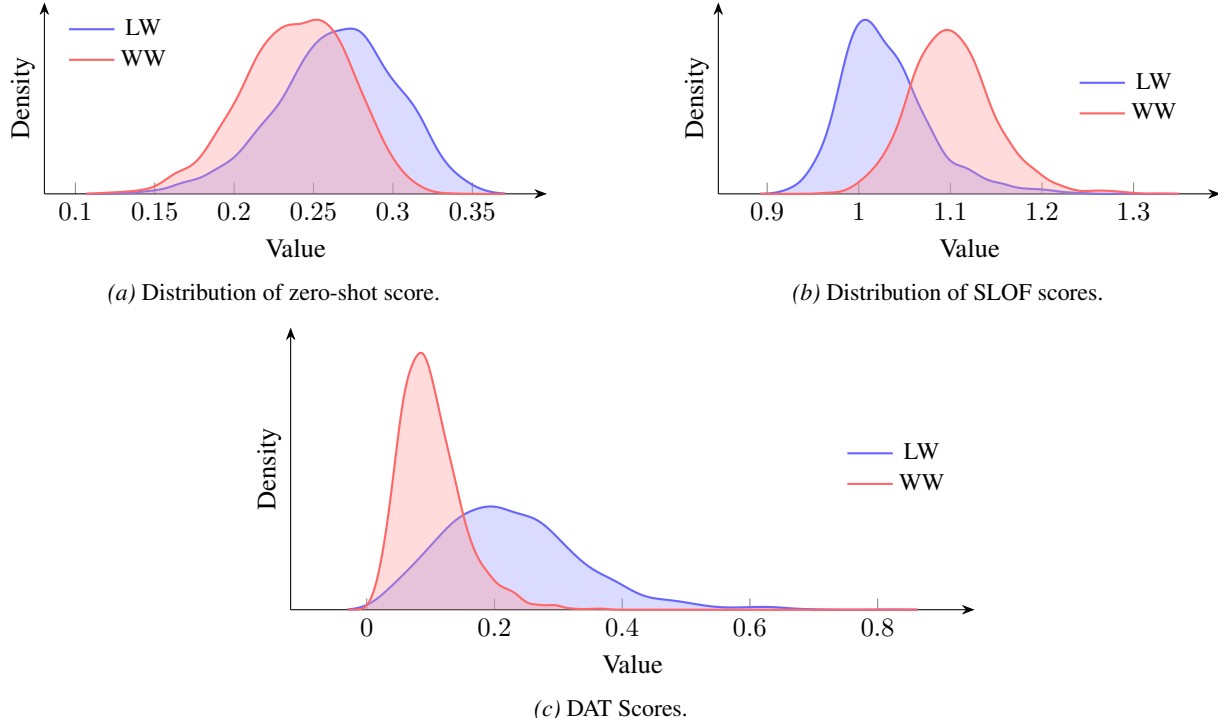

*(a)* Distribution of zero-shot score.

*(b)* Distribution of SLOF scores.

*(c)* DAT Scores.

*Figure 6.* Illustration of score distributions for the worst group in Waterbirds (Landbird in Water, LW) and its spuriously aligned counterpart (Waterbird in Water, WW). (a) Zero-shot cosine scores for both groups largely overlap, making discrimination difficult. (b) SLOF values show clearer separation, with LW having lower SLOF under its own group references. (c) DAT scores incorporate this density difference and sharpen the separation.

although global normalisation can partially reduce bias, it does not address the subgroup-level geometric imbalance that DAT is designed to correct.

*Table 9.* Comparison with global normalisation baselines on Waterbirds. MC denotes mean-centering and W denotes whitening. DAT achieves the highest worst-group accuracy across backbones.

| Method | CLIP (ViT-B/32) | | | CLIP (ViT-L/14) | | | CLIP (ResNet50) | | |
|---|---|---|---|---|---|---|---|---|---|
| | WG | Avg | Gap | WG | Avg | Gap | WG | Avg | Gap |
| ZS | 41.37 | 68.48 | 27.11 | 31.93 | 83.72 | 51.79 | 35.36 | 80.64 | 45.28 |
| ZS+MC | 47.19 | 70.52 | 23.33 | 53.66 | 75.89 | 22.23 | 45.76 | 70.12 | 24.36 |
| ZS+W | 45.59 | 51.86 | 6.27 | 47.20 | 52.31 | 5.11 | 51.53 | 53.07 | 1.54 |
| Group Prompt | 43.46 | 68.48 | 27.11 | 31.93 | 83.72 | 51.79 | 35.36 | 80.64 | 45.28 |
| Group Prompt+MC | 58.88 | 78.82 | 29.94 | 66.51 | 83.03 | 16.52 | 58.41 | 78.18 | 19.77 |
| Group Prompt+W | 52.24 | 55.01 | 2.77 | 47.66 | 52.33 | 4.67 | 51.04 | 53.97 | 2.93 |
| DAT | **75.08** | **80.36** | 5.28 | **83.33** | **89.57** | 6.42 | **75.08** | **83.83** | 8.75 |

## C.3. Sensitivity to Reference-Set Quality

We further evaluate the sensitivity of DAT to the quality of the reference set. In particular, we corrupt the group assignments of reference samples to simulate mislabeled or noisy group references, and evaluate DAT under different noise levels. As shown in Table 10, DAT remains stable when the noise level increases from 0% to 15%. The worst-group accuracy changes only slightly across backbones, and the average accuracy also shows minor fluctuations. Although the gap varies at some noise levels, there is no consistent degradation trend as noise increases. These results indicate that DAT does not strongly depend on perfectly clean group references and can tolerate moderate reference-set noise.

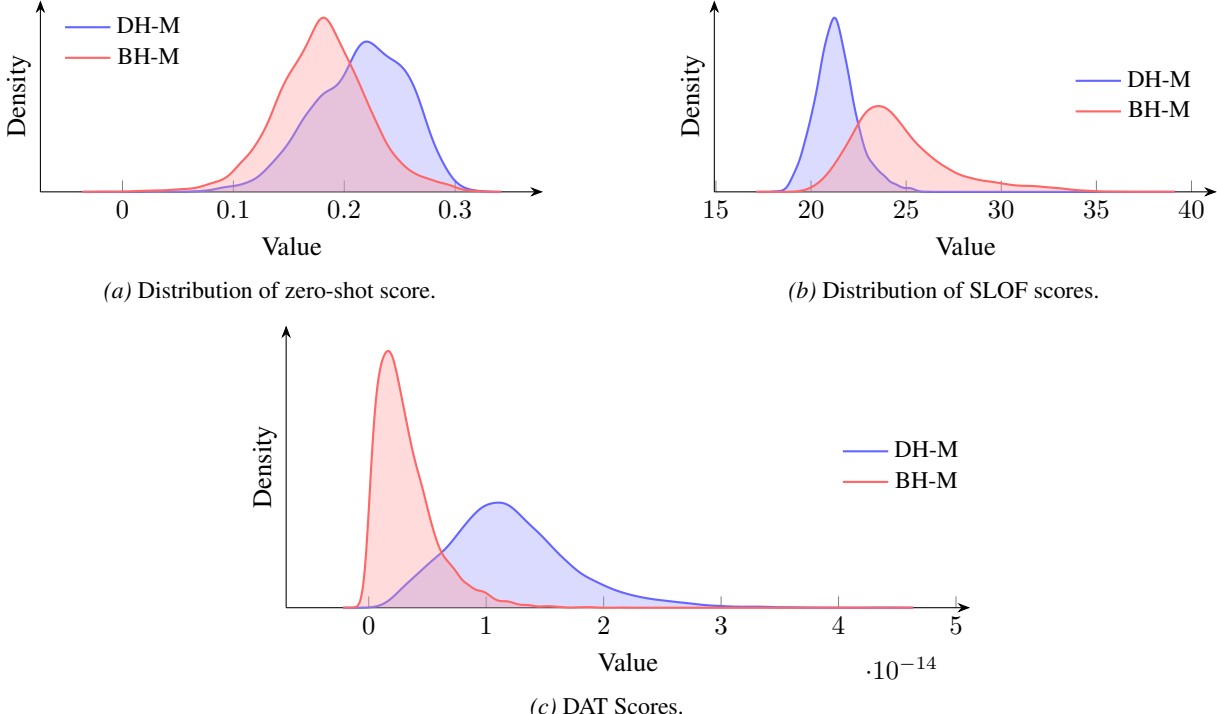

*(a)* Distribution of zero-shot score.

*(b)* Distribution of SLOF scores.

*(c)* DAT Scores.

*Figure 7.* Illustration of score distributions for Dark-Hair Male (DH-M) in CelebA, and its spuriously aligned counterpart (Blond-Hair Male, BH-M). (a) Zero-shot cosine scores for both groups largely overlap, making discrimination difficult. (b) SLOF values show clearer separation, with DH-M having lower SLOF under its own group references. (c) DAT scores incorporate this density difference and sharpen the separation.

*Table 10.* Sensitivity to reference-set quality on Waterbirds. DAT remains stable under moderate group-assignment noise in the reference set.

| Noise | CLIP (ViT-B/32) | | | CLIP (ViT-L/14) | | | CLIP (ResNet50) | | |
|---|---|---|---|---|---|---|---|---|---|
| | WG | Avg | Gap | WG | Avg | Gap | WG | Avg | Gap |
| 0% | 75.08 | 80.36 | 5.28 | 83.33 | 89.57 | 6.42 | 75.08 | 83.83 | 8.75 |
| 5% | 74.14 | 81.48 | 7.34 | 80.37 | 90.27 | 9.90 | 74.77 | 83.38 | 8.61 |
| 10% | 74.61 | 80.67 | 6.06 | 83.49 | 89.45 | 5.96 | 74.61 | 86.81 | 12.20 |
| 15% | 76.01 | 81.62 | 5.61 | 83.65 | 89.37 | 5.72 | 74.29 | 83.79 | 9.50 |

## D. Robust Text Prompt

### D.1. Robust Text Prompt Details - Waterbirds

To evaluate group-robust text prompts, we follow Lu et al. (2025) and adopt their evaluated sentences, generated with GPT-4 (OpenAI, 2023). For Waterbirds, we generate robustified prompts by creating multiple variants of land and water descriptions to represent the spurious attribute (Table 11), along with corresponding group prompts (Table 12).

| Land | Water |
|---|---|
| A photo of a land background | A photo of a water background |
| A photo of a forest background | A photo of an ocean background |
| A photo of a mountain background | A photo of a sea background |
| A photo of a terrain background | A photo of a lake background |
| A photo of a ground background | A photo of a river background |

*Table 11.* Spurious prompts used for robust prompt prompt evaluation of Waterbirds dataset (Lu et al., 2025).

| Landbird - Land | Waterbird - Water |
|---|---|
| A photo of a landbird in land | A photo of a waterbird in water |
| A photo of a landbird in forest | A photo of waterbird in ocean |
| A photo of a landbird in mountain | A photo of a waterbird in sea |
| A photo of a landbird in terrain | A photo of a waterbird in lake |
| A photo of a landbird in ground | A photo of a waterbird in river |

| Landbird - Water | Waterbird - Land |
|---|---|
| A photo of a landbird in water | A photo of a waterbird in land |
| A photo of a landbird in ocean | A photo of waterbird in forest |
| A photo of a landbird in sea | A photo of a waterbird in mountain |
| A photo of a landbird in lake | A photo of a waterbird in terrain |
| A photo of a landbird in river | A photo of a waterbird in ground |

*Table 12.* Group prompts used for robust prompt evaluation of Waterbirds dataset.

## D.2. Robust Text Prompt for CelebA

We further evaluate the robustness of DAT and DAT* to prompt variability on the CelebA dataset. For this setting, we construct group-aligned prompts by generating multiple variants of the spurious attribute using GPT-4 (OpenAI, 2023), as shown in Table 15. We also evaluate group prompts based on female and male variants (Table 16). These variants are averaged to form robust text embeddings. As reported in Table 13, DAT behaves similarly, with only minor changes in WG and Avg accuracy. On the other hand, Table 14 shows DAT* improvement, as prompt semantics influence the inferred spurious attribute and its alignment to image embeddings.

*Table 13.* Group robustify prompting evaluation for DAT on the CelebA.

| Method | ViT-B/32 | | | ViT-L/14 | | | ResNet-50 | | |
|---|---|---|---|---|---|---|---|---|---|
| | WG | Avg | Gap | WG | Avg | Gap | WG | Avg | Gap |
| DAT | 78.53 | 87.09 | 8.56 | 85.35 | 86.54 | 1.19 | 80.79 | 87.09 | 6.30 |
| DAT Robust | 78.53 | 87.08 | 8.55 | 85.05 | 86.60 | 1.55 | 81.36 | 86.62 | 5.26 |

*Table 14.* Group robustify prompting evaluation for DAT* on the CelebA.

| Method | ViT-B/32 | | | ViT-L/14 | | | ResNet-50 | | |
|---|---|---|---|---|---|---|---|---|---|
| | WG | Avg | Gap | WG | Avg | Gap | WG | Avg | Gap |
| DAT* | 78.53 | 87.11 | 8.58 | 84.93 | 86.54 | 1.61 | 78.53 | 88.29 | 9.76 |
| DAT* Robust | 79.10 | 87.20 | 8.10 | 85.06 | 86.61 | 1.55 | 81.92 | 86.85 | 4.93 |

| Female | Male |
|---|---|
| A photo of a female | A photo of a male |
| A photo of a woman | A photo of a man |
| A photo of a lady | A photo of a gentleman |
| A photo of a girl | A photo of a boy |

*Table 15.* Spurious prompts used for robust prompt prompt evaluation of the CelebA dataset.

| Black Hair - Female | Blonde Hair - Male |
|---|---|
| `a photo of a female celebrity with dark hair` | `a photo of a male celebrity with blonde hair` |
| `a photo of a woman celebrity with dark hair` | `a photo of a man celebrity with blonde hair` |
| `a photo of a lady celebrity with dark hair` | `a photo of a gentleman celebrity with blonde hair` |
| `a photo of a girl celebrity with dark hair` | `a photo of a boy celebrity with blonde hair` |

| Black Hair - Male | Blonde Hair - Female |
|---|---|
| `a photo of a female celebrity with dark hair` | `a photo of a male celebrity with blonde hair` |
| `a photo of a female celebrity with dark hair` | `a photo of a woman celebrity with blonde hair` |
| `a photo of a gentleman celebrity with dark hair` | `a photo of a lady celebrity with blonde hair` |
| `a photo of a boy celebrity with dark hair` | `a photo of a girl celebrity with blonde hair` |

*Table 16.* Group prompts used for robust prompt evaluation of the CelebA dataset.

# E. Discussion on Text Prompts

## E.1. Different Spurious Text Prompt Templates

In addition to the specific wording of the spurious feature itself, the structure of the prompt template can also influence performance. To examine this effect more thoroughly, we evaluated all methods under two template formats, using T1: "{spurious feature label}", and T2: "A photo with a spurious feature, {spurious feature label}" on the Waterbirds dataset and CLIP ViT-B/32. We exclude TIE and DAT from this analysis because they assume access to spurious labels and therefore do not respond to changes in spurious textual formulation. As shown in Table 17, performance remains stable across both formats for DAT*, which achieves the highest worst-group and average accuracy. This suggests that DAT* is robust to prompt phrasing.

*Table 17.* Effect of prompt template variation on Waterbirds with CLIP ViT-B/32.

| Method | T1 Spurious Template | | | T2 Spurious Template | | |
|---|---|---|---|---|---|---|
| | WG | Avg | Gap | WG | Avg | Gap |
| ZS | 41.37 | 64.48 | 27.11 | 41.37 | 64.48 | 27.11 |
| Group Prompt | 43.46 | 66.79 | 23.33 | 43.46 | 66.79 | 23.33 |
| Ideal Words | 61.99 | 78.87 | **16.88** | 60.28 | 79.20 | 18.92 |
| PerceptionCLIP | 23.37 | 61.54 | 38.17 | 59.78 | **82.50** | 22.72 |
| ROBOSHOT | 44.35 | 69.03 | 24.68 | 54.41 | 71.92 | 17.51 |
| TIE* | 56.14 | 75.00 | 18.86 | 61.24 | 76.91 | **15.67** |
| DAT* | **64.49** | **82.74** | 18.25 | **64.02** | 82.33 | 18.31 |

## E.2. Text Prompts Effect Evaluation

Identifying strategies to build reliable and generalisable prompts is an open challenge. To examine this further, we conducted a series of experiments evaluating how different prompt formats and levels of object specificity affect DAT performance. To investigate prompt design, we decompose each text prompt into two parts: a *template* and an *object term*, following prior work Lu et al. (2025). We apply the following templates for prompts:

- T1: A photo with a `[Object]` background

- T2: A photo with a spurious feature, `[Object]`

- T3: `[Object]`

Building on the findings of Ge et al. (2023), which show that object labels exhibit a semantic hierarchy as captured in WordNet (Fellbaum, 1998), we investigate three strategies for selecting object terms in our study: (i) using the immediate parent node in the hierarchy to represent a broader category, (ii) using the spurious feature itself, and (iii) averaging the embeddings of five highly specific sibling terms from the bottom of the hierarchy to capture fine-grained detail. Table 18

lists the candidate terms. This setup allows us to evaluate which level of abstraction yields the most effective prompts. This design allows us to probe how the specificity or abstraction level of the object term impacts model predictions. We conduct these experiments using the DAT* method with CLIP ViT-L/14, which demonstrated strong baseline performance.

*Table 18.* Object term variants based on WordNet hierarchy.

| Granularity | Water Background | Land Background |
|---|---|---|
| O1 (Hypernyms) | fluid | ground |
| O2 (Original) | water | land |
| O3 (Hyponyms) | sea water, lake water, | farmland, forest land, |
| | river water, stream water, creek water | arable land, grassland, desert land |

*Table 19.* Prompt structure ablation on Waterbirds with CLIP ViT-L/14 using DAT*.

| Prompt | WG | Avg | Gap |
|---|---|---|---|
| T1 + O1 | 67.44 | 87.02 | 19.58 |
| T1 + O2 | 79.75 | 87.87 | 8.12 |
| T1 + O3 | 80.09 | 85.76 | 5.67 |
| T2 + O1 | 67.60 | 86.00 | 18.04 |
| T2 + O2 | 79.91 | 86.95 | 7.04 |
| T2 + O3 | 80.18 | 85.47 | 5.29 |
| T3 + O1 | 68.69 | 85.90 | 17.21 |
| T3 + O2 | 82.09 | 87.81 | 5.72 |
| T3 + O3 | 80.37 | 89.06 | 8.69 |

From Table 19, we observe that prompt phrasing (T1–T3) has a modest influence, with DAT* demonstrating resilience to syntactic variations. However, the object term plays a larger role: fine-grained and semantically aligned terms (O3) consistently cause higher WG accuracy, while overly broad descriptors like hypernyms (O1) degrade performance. These findings reinforce the importance of context-aware and precise descriptions for zero-shot robustness.

