# OpenReview forum: "Density-Aware Translation of Spurious Correlations in Zero-Shot VLMs"
_ICML.cc/2026/Conference — ICML 2026 regular_

### Official Review · Reviewer_C3Ku · 2026-03-09

**Soundness:** 2
**Presentation:** 3
**Significance:** 2
**Originality:** 2
**Overall Recommendation:** 4
**Confidence:** 4

**Summary:**

This paper addresses spurious correlations in zero-shot vision-language models (VLMs). It identifies that anisotropic (elliptical) embeddings cause cosine similarity to overvalue samples aligned with spurious attributes. To counter this, the authors propose Density-Aware Translation (DAT), a training-free post-processing method that rescales image-text similarities using a local density proxy (SLOF) computed from small group-level reference sets. A variant, DAT*, infers spurious attributes via zero-shot classification when labels are unavailable. Experiments on Waterbirds, CelebA, COVID-19, and FMoW show improvements in worst-group accuracy across multiple VLMs. Theoretical analysis using the Kent distribution is provided.

**Compliance With Llm Reviewing Policy:**

Affirmed.

**Final Justification:**

The rebuttal has solved my concerns, and I recommend the paper week accept.

**Key Questions For Authors:**

1.	The main DAT method requires spurious attribute labels for reference set construction. In a true zero-shot setting where only class names are provided, how would a practitioner obtain these labels? Doesn't this violate the zero-shot assumption?
2.	DAT* relies on zero-shot attribute inference. How robust is it to errors in this step? Is there a correlation between attribute prediction accuracy and final worst-group gain?
3.	For datasets with many groups (e.g., FMoW with 310 groups), maintaining n samples per group becomes cumbersome. How would you recommend applying DAT in such large-group settings?

**Limitations:**

Yes.

**Strengths And Weaknesses:**

Strengths:
1.	DAT is a lightweight, training-free post-processing method requiring only small reference sets, making it easy to integrate.
2.	The method consistently improves worst-group accuracy across diverse datasets and multiple VLM backbones.
3.	Linking cosine bias to the anisotropic geometry of CLIP embeddings is a valuable and well-motivated observation.
4.	The use of the Kent distribution to model anisotropy and prove DAT's correction is mathematically sound.
Weaknesses:
1.	The core DAT method requires both class labels and spurious attribute labels to construct group-wise reference sets. This is a significant assumption that is often unrealistic in zero-shot deployment. DAT* relaxes this but becomes critically dependent on the accuracy of zero-shot attribute inference, which is itself unreliable.
2.	Using pre-constructed, group-labeled reference sets constitutes a form of target-domain supervision. This blurs the definition of "zero-shot" and weakens the paper's core claim.
3.	The technical components (SLOF, herding) are borrowed from existing literature. The contribution is the insightful application of these tools, not a novel algorithmic advance.
4.	DAT introduces several hyperparameters (k, n, λ) that require tuning per dataset, reducing its "plug-and-play" appeal.

---

> ### Author Rebuttal · Authors · 2026-03-31
>
> We sincerely appreciate the reviewer’s time and valuable feedback. We would like to clarify the following points.
>
> **W1 & W2 (Access to labels / Dependence on attribute classification accuracy).**
>
> **A-W1 & W2:** Following prior work [1], our setting follows standard zero-shot learning [2], where no training, fine-tuning, or target labels are used to update model parameters. The reference set in DAT is only used to estimate representation geometry and does not involve any training or adaptation of the model. Using external or auxiliary labeled data for calibration without updating model weights is common practice in zero-shot evaluation and does not violate the zero-shot assumption.
>
> We also note that DAT does not strictly require ground-truth spurious attributes. DAT* relaxes this by estimating attributes via zero-shot inference. Despite prediction errors, DAT* achieves performance comparable to DAT, showing robustness to attribute noise. As shown in response to Q2 of Reviewer 4Vkd, even with low attribute prediction accuracy (e.g., ViT-L/14), DAT* still achieves strong performance.
>
> **W3 (Technical components from existing literature).**
>
> **A-W3:** Our contribution is the novel integration of components to both empirically and theoretically address a previously underexplored problem of how subgroup density differences in VLM representation space affect worst-group performance in zero-shot settings.
>
> To our knowledge, this is the first work to (1) identify density imbalance as a key geometric factor behind spurious correlations in zero-shot VLMs, (2) show how density-aware score correction can mitigate this effect without model retraining, and (3) provide both theoretical and empirical analysis connecting representation geometry to worst-group robustness. While SLOF and herding are known tools, their use in this context and their combination into a training-free correction framework are new.
>
> Also, DAT provides a principled correction to cosine similarity bias by incorporating density information that captures anisotropic structure ignored by cosine. This makes DAT both theoretically grounded and effective, while remaining computationally efficient (Table 7).
>
> **W4 (Plug-and-play in DAT).**
>
> **A-W4:** We verified through the ablation study in Section 4 and Figure 3 (a–c) that DAT maintains strong performance across a wide range of density-related hyperparameters, including the scaling factor $\lambda$, the number of neighbours $k$, and the reference-set size $n$. Results show DAT is not sensitive to precise tuning and supports a plug-and-play setup.
>
>
> **Q1 (Zero-shot setting).**
>
> **A-Q1:** Our setting remains consistent with the standard definition of zero-shot learning [2], where the model is evaluated on a target dataset without any task-specific training or access to its labels. The reference set is only used to estimate representation geometry and does not involve any fine-tuning or supervision from the target-domain data. Using external images for this purpose is still considered zero-shot, as long as the method does not use labeled samples from the evaluation dataset, which is the case in our setup.
>
> In addition, in DAT*, these attributes can also be obtained via zero-shot inference. Our experiments in Tables 1 and 2 show that DAT* achieves performance comparable to DAT, demonstrating that the method remains applicable even when spurious attributes are not available. Therefore, DAT can be applied in practical zero-shot scenarios either using external unlabeled data or automatically inferred attributes without violating the zero-shot assumption.
>
> **Q2 (Robustness to the errors of attribute prediction).**
>
> **A-Q2:** Due to the character limit, we kindly refer the reviewer to our response to Q2 of Reviewer 4Vkd.
>
> **Q3 (DAT for large-group).**
>
> **A-Q3:** As shown in Algorithms 1 and 2, when a group’s reference set contains fewer than n samples, we set its density to $\infty$, which forces the corresponding corrected score to zero. In practice, this causes the model to automatically skip unreliable subgroups and rely on the remaining groups for prediction. This behavior appears in FMoW and contributes to DAT’s robustness in large-scale settings. Because the reference sets are small and the fallback mechanism handles missing or underrepresented groups, DAT scales efficiently without requiring full group coverage or large labeled datasets.
>
> [1] Lu, Shenyu, Junyi Chai, and Xiaoqian Wang. "Mitigating Spurious Correlations in Zero-Shot Multimodal Models." International Conference on Learning Representations (ICLR), 2025.
>
> [2] Wang, Wei, et al. "A survey of zero-shot learning: Settings, methods, and applications." ACM Transactions on Intelligent Systems and Technology (TIST) 10.2 (2019): 1-37.

---

> > ### Author Rebuttal · Reviewer_C3Ku · 2026-04-03
> >
> > The rebuttal has solved my concerns,  and I recommend the paper week accept.

---

> > > ### Author Response · Authors · 2026-04-04
> > >
> > > Thank you so much for your time and for carefully considering our rebuttal. We are glad that our clarifications addressed your concerns.
> > >
> > > We noticed that the updated recommendation has not yet appeared on OpenReview. Would you please be able to update your recommendation, or is there anything else you require?

---

### Official Review · Reviewer_QKv3 · 2026-03-12

**Soundness:** 3
**Presentation:** 3
**Significance:** 3
**Originality:** 3
**Overall Recommendation:** 4
**Confidence:** 2

**Summary:**

Density-Aware Translation (DAT) addresses the susceptibility of zero-shot vision–language models to spurious correlations by incorporating local geometric density information into image–text similarity scoring. The method is motivated by the observation that CLIP embeddings exhibit an anisotropic ellipsoidal structure, where frequently occurring but potentially spurious patterns cluster near the mean while rare yet semantically meaningful samples lie in sparser regions of the embedding space. DAT constructs small group-specific reference sets and estimates local density using a nearest-neighbour–based proxy, then rescales similarity scores to penalize matches occurring in diffuse regions while preserving dense and semantically consistent alignments. This correction can be interpreted theoretically as reinstating log-likelihood terms missing from cosine similarity under anisotropic distributions. Extensive zero-shot experiments across multiple datasets and vision–language models demonstrate that DAT consistently improves worst-group and average accuracy while reducing the performance gap across groups, without requiring model fine-tuning or additional supervision.

**Compliance With Llm Reviewing Policy:**

Affirmed.

**Key Questions For Authors:**

Despite its strengths, the work has several limitations.

First, the novelty of the method may be somewhat limited. The core idea of adjusting similarity scores using density estimates can be viewed as a relatively straightforward extension of existing embedding calibration techniques. As such, the conceptual contribution may appear incremental compared to prior work on zero-shot debiasing.

Second, the approach relies on the construction of group-specific reference sets, which may limit scalability in settings with a large number of classes or attributes. The need to estimate local densities for each group may also introduce additional computational overhead during inference.

Third, the method assumes the availability of meaningful attribute prompts or the ability to infer attributes via zero-shot classification. In practice, defining appropriate attribute prompts may require domain knowledge, which could limit the applicability of the approach in more open-ended settings.

**Limitations:**

see in questions

**Strengths And Weaknesses:**

Soundness

The paper is generally technically sound. The proposed Density-Aware Translation (DAT) method is conceptually simple and well motivated by the geometric properties of CLIP embeddings. The authors provide a theoretical analysis showing that cosine similarity may be biased under anisotropic embedding distributions and argue that DAT approximately restores a density-aware log-likelihood term in the decision rule. While the theoretical derivation relies on assumptions linking the SLOF density estimator to the underlying data distribution, these assumptions are standard in density estimation literature and appear reasonable within the scope of the analysis.

Empirically, the method is evaluated across several widely used spurious correlation benchmarks, including Waterbirds, CelebA, COVID-19, and FMoW, using multiple VLM backbones. The experiments report improvements in both worst-group accuracy and average accuracy compared to prior approaches. The evaluation protocol follows common practices in the literature and includes ablation studies examining the sensitivity to hyperparameters such as the number of reference samples, the scaling factor, and neighborhood size. Overall, the empirical results support the main claims of the paper, although additional experiments could further strengthen the robustness of the conclusions.

Presentation

The paper is clearly written and generally well structured. The motivation is clearly introduced through the geometric analysis of CLIP embeddings, and the figures illustrating the anisotropic structure of the embedding space help convey the intuition behind the method. The pipeline description of DAT and DAT* is straightforward and easy to follow.

The related work section provides a reasonable overview of existing approaches for mitigating spurious correlations in vision-language models, including both fine-tuning-based and zero-shot approaches. The paper also clearly distinguishes the proposed approach from methods that rely on prompt engineering or embedding projections.

One area that could be improved is the clarity of the methodological exposition. Some parts of the density estimation and reference set construction could benefit from additional intuition or implementation details to facilitate reproducibility. In particular, the sampling strategy for constructing reference sets and the practical computational overhead of density estimation could be discussed more explicitly.

Significance

The paper addresses an important problem in multimodal machine learning: the vulnerability of vision-language models to spurious correlations in zero-shot settings. As VLMs such as CLIP are increasingly used in real-world applications, improving their robustness to spurious correlations is both practically and scientifically relevant.

The proposed approach has several appealing practical properties. It operates in a purely zero-shot setting, does not require fine-tuning the model, and can be applied as a lightweight post-processing step. This makes the method potentially useful in scenarios where access to training data or model parameters is limited.

While the empirical improvements are moderate in some settings, the approach could be valuable as a calibration mechanism that improves group robustness without modifying the underlying model. Therefore, the work has potential practical relevance for applications involving fairness, domain shift, or distributional robustness.

Originality

The main novelty of the paper lies in incorporating local geometric density information into image–text similarity scoring for zero-shot classification. While density-based ideas and kNN-based statistics are well studied in representation analysis, their use as a calibration mechanism for vision-language similarity scores appears to be a relatively new perspective.

The work does not introduce a fundamentally new learning framework but rather provides a principled modification of the similarity scoring function. The originality therefore stems more from the geometric interpretation and the integration of density-aware corrections into the zero-shot inference pipeline than from algorithmic complexity.

The theoretical analysis linking anisotropic embeddings to biased cosine similarity provides additional insight into the behavior of CLIP-like models. Although the theoretical framework relies on standard assumptions, it helps explain why density-aware corrections may improve robustness.

Weaknesses

Despite its strengths, the work has several limitations.

First, the novelty of the method may be somewhat limited. The core idea of adjusting similarity scores using density estimates can be viewed as a relatively straightforward extension of existing embedding calibration techniques. As such, the conceptual contribution may appear incremental compared to prior work on zero-shot debiasing.

Second, the approach relies on the construction of group-specific reference sets, which may limit scalability in settings with a large number of classes or attributes. The need to estimate local densities for each group may also introduce additional computational overhead during inference.

Third, the method assumes the availability of meaningful attribute prompts or the ability to infer attributes via zero-shot classification. In practice, defining appropriate attribute prompts may require domain knowledge, which could limit the applicability of the approach in more open-ended settings.

---

> ### Author Rebuttal · Authors · 2026-03-31
>
> Thank you for taking the time to review our paper and providing valuable feedback. Below, you will find our responses to your questions:
>
> **Q1 (Idea and Contribution).**
>
> **A-Q1:** Unlike prior zero-shot debiasing methods that mainly rely on prompt engineering or logit adjustments, DAT introduces a representation-level correction based on local density structure. This provides a new perspective on why spurious correlations arise in VLMs and how they can be mitigated through geometric correction rather than model modification.
>
> To our knowledge, this is the first work to (1) identify subgroup density imbalance as a key geometric factor underlying spurious correlations in zero-shot VLMs, (2) show that density-aware score correction can mitigate this effect without any model retraining or access to target labels, and (3) provide both theoretical and empirical analysis linking representation geometry to worst-group robustness. The novelty and significance of our contributions are also reflected in the other reviewer's feedback. Reviewer 4Vkd highlighted the geometric interpretation of spurious correlations, the practical advantages of our post-hoc design, and the theoretical grounding of DAT through anisotropic density modeling. Reviewer JvEj also noted the theoretical justification of our approach under feature density assumptions, further supporting the soundness of our method.
>
> **Q2 (Handling a large number of classes and attributes/ Computational overhead).**
>
> **A-Q2:** We run experiments in Section 4.1 (Table 3-b) to evaluate the scalability of that DAT/DAT*, demonstrating that the method remains effective on datasets with a large number of classes and attributes. For example, the FMoW dataset contains 62 classes and 5 spurious attributes, resulting in 310 subgroups. As shown in Table 3, DAT remains effective without requiring any modification to the method for the FMoW dataset. This demonstrates that the approach is not limited to small subgroup settings.
>
> Regarding computational cost, we also report runtime comparisons in Table 7 (Appendix B.5), where we showed that the translation step requires less than about 200 seconds on an NVIDIA H100 GPU. In comparison, TIE/TIE*, the most recent zero-shot baselines, take approximately more than 900 seconds under the same hardware, settings, and dataset, which is more than four times the latency of DAT/DAT*, demonstrating that our approach is significantly more efficient and therefore better suited for large-scale or time-sensitive deployment.
>
> **Q3 (Access to attribute prompts).**
>
> **A-Q3:** This assumption is consistent with prior work on zero-shot debiasing and robustness in VLMs, such as TIE/TIE* [1], ROBOSHOT [2], and PerceptionCLIP [3], which also rely on defining attribute or context prompts. Similarly, DAT assumes access to reasonable attribute prompts, which is a standard and practical assumption in zero-shot multimodal robustness settings.
>
> Moreover, as discussed in Appendices D.2 and E, our method is not highly sensitive to the exact prompt template or the specific formulation of the attribute descriptions. Our experiments show that variations in attribute phrasing or prompt design have only a minor impact on performance, indicating that DAT does not require carefully engineered prompts to be effective.
>
> [1] Lu, Shenyu, Junyi Chai, and Xiaoqian Wang. "Mitigating Spurious Correlations in Zero-Shot Multimodal Models." International Conference on Learning Representations (ICLR), 2025.
>
> [2] Adila, Dyah, et al. "Zero-Shot Robustification of Zero-Shot Models." International Conference on Learning Representations (ICLR), 2024.
>
> [3] An, Bang, et al. "PerceptionCLIP: Visual Classification by Inferring and Conditioning on Contexts." International Conference on Learning Representations (ICLR), 2024.

---

> > ### Author Rebuttal · Reviewer_QKv3 · 2026-04-08
> >
> > Thank you for the author's reply. I will maintain the score.

---

### Official Review · Reviewer_JvEj · 2026-03-12

**Soundness:** 3
**Presentation:** 3
**Significance:** 3
**Originality:** 3
**Overall Recommendation:** 5
**Confidence:** 3

**Summary:**

Vision-language models such as CLIP exhibit spurious correlations and semantic blurring within their representations. This leads to poor performance for zero-shot learning using cosine similarity on datasets with similar contextual cues.  The authors propose a method to rescale image-prompt similarities by the estimated relative density of the query image representation within each group. Relative density is estimated using SLOF on N representative image samples for each group.

**Compliance With Llm Reviewing Policy:**

Affirmed.

**Key Questions For Authors:**

1) I believe "group prompt" is equivalent to s_{y,a}. Did you compare performance using just \tilde{s}_{y,a}  and just \tilde{s}_{y,avg} ?
2) In Figure 1b, Tangent-space Mahalanobis distance is used, but I would have liked to also see  cosine similarity. According to the figure, "landbird on land" photos actually have a smaller distance to the prompt "a photo of a waterbird" than "waterbird on land" photos. If the photos are conditioned on land, then landbirds (on average) are actually closer to the waterbird prompt than the waterbirds. Is that the case for cosine distance? If not, does that suggest tangent-space Mahalanobis distance isn't working well in this case?
3) You mention the modality gap. Did you compare results using simple techniques like mean-centering or whitening features by modality?

**Limitations:**

yes

**Strengths And Weaknesses:**

Soundness:
The claims in the paper are well supported by experiment on multiple datasets. The approach is theoretically justified under the certain feature density assumptions. The results are compared with a variety of approaches. The authors also compare how other density estimation methods affect the approach.

Presentation:
The paper is written clearly. The authors could have been more explicit in explaining the reasoning for each choice made in 3.3.

Weaknesses:
DAT requires there be representative examples for each group, and each group be known ahead of time. DAT* does not require groups be labeled but does not perform as well.

---

> ### Author Rebuttal · Authors · 2026-03-31
>
> We sincerely thank the reviewer for the careful and constructive feedback. We will incorporate these clarifications and the additional experimental results in the final revision.
>
> **Q1 (Using different terms of aggregation).**
>
> **A-Q1:** Yes, the group prompt corresponds to the group-specific corrected scores $\tilde{s}_{y,a}​$. As shown in the table below, although using either $\tilde{s}{y,a}$ or $\tilde{s}{y,\mathrm{Avg}}$ alone improves performance compared to the baselines, neither alone matches the full DAT formulation. Using only $\tilde{s}{y,a}$ improves subgroup robustness by correcting density imbalance, but may slightly reduce stability since predictions rely only on subgroup-specific corrections (High Gap). In contrast, $\tilde{s}{y,\mathrm{Avg}}$ provides more balanced performance by aggregating information across groups, but does not fully exploit subgroup-specific corrections. Their combination in DAT achieves the best worst-group accuracy while maintaining strong average performance, showing that the two components play complementary roles for optimal performance.
>
> ||$\mathrm{DAT}$|||$\tilde{s}_{y,a}$|||$\tilde{s}_{y,avg}$|||
> |---|---|---|---|---|---|---|---|---|---|
> | |WG|Avg|Gap|WG|Avg|Gap|WG|Avg|Gap|
> |CLIP (ViT-B/32)|75.08|80.36|5.28|70.56|80.49|9.93|72.94|81.43|8.49|
> |CLIP (ViT-L/14)|83.33|89.57|6.42|74.14|89.73|15.59|81.46|89.24|7.78|
> |CLIP (ResNet50)|75.08|83.83|8.75|77.78|83.06|5.28|74.36|83.69|9.33|
>
> **Q2 (Tangent Mahalanobis vs cosine).**
>
> **A-Q2:** We used tangent-space Mahalanobis distance because it captures the local covariance structure of each subgroup, whereas cosine distance reflects only global angular alignment.
>
> To address your question and verify whether the observation in Figure 1(b) is specific to tangent Mah. distance, we computed cosine distances between subgroup means and the prompt “a photo of a waterbird.” The results are shown below.
>
> |subgroup|Tan. Mah.|Cos.|
> |---|---|---|
> |LL|36.97|0.702|
> |LW|36.81|0.652|
> |WL|37.82|0.668|
> |WW|35.48|0.643|
>
> We observe that both metrics produce a consistent qualitative ordering. The WW group shows the smallest distance under both measures, which is expected since both the class label and the spurious attribute align with the waterbird prompt. Conversely, the LL and WL groups exhibit the largest distances under both metrics, which is also consistent since these samples share the land background that conflicts with the waterbird context.
>
> Importantly, the LW group shows a smaller distance than the WL group under both metrics. This indicates that the effect observed in Figure 1(b) is not specific to the tangent Mahalanobis distance but is already present when using cosine similarity.
>
>  **Q3 (Mean-centering and Whitening).**
>
> **A-Q3:** Thank you for this suggestion. Mean-centering (MC) and Whitening (W) can be useful for reducing global distribution differences between image and text embeddings when sufficient samples from both modalities are available. DAT addresses a different issue, it targets local density differences between subgroups. The effect shown in Figure 1(b) arises from subgroup geometry rather than global modality misalignment, so global normalization alone cannot change this relative structure.
>
> We experimentally evaluated MC and W. As shown in the table below, MC provides moderate improvements over zero-shot and group prompts, indicating that some global bias exists. W does not consistently improve performance and sometimes reduces average accuracy, which is expected since it removes global anisotropy that may also contain useful semantic structure. In contrast, DAT consistently achieves the best worst-group accuracy across all backbones. These results suggest that while global normalization can partially reduce bias, it does not address subgroup-level geometric imbalance, which DAT is specifically designed to handle.
>
> | |CLIP (ViT-B/32)|||CLIP (ViT-L/14)|||CLIP (ResNet50)|||
> |---|---|---|---|---|---|---|---|---|---|
> | |WG|Avg|Gap|WG|Avg|Gap|WG|Avg|Gap|
> |ZS|41.37|68.48|27.11|31.93|83.72|51.79|35.36|80.64|45.28|
> |ZS+MC|47.19|70.52|23.33|53.66|75.89|22.23|45.76|70.12|24.36|
> |ZS+W|45.59|51.86|6.27|47.20|52.31|5.11|51.53|53.07|1.54|
> |Group Prompt|43.46|68.48|27.11|31.93|83.72|51.79|35.36|80.64|45.28|
> |Group Prompt+MC|58.88|78.82|29.94|66.51|83.03|16.52|58.41|78.18|19.77|
> |Group Prompt+W|52.24|55.01|2.77|47.66|52.33|4.67|51.04|53.97|2.93|
> |DAT|75.08|80.36|5.28|83.33|89.57|6.42|75.08|83.83|8.75|

---

> > ### Author Rebuttal · Reviewer_JvEj · 2026-04-03
> >
> > The authors have addressed my questions and concerns.

---

### Official Review · Reviewer_4Vkd · 2026-03-15

**Soundness:** 2
**Presentation:** 2
**Significance:** 2
**Originality:** 2
**Overall Recommendation:** 4
**Confidence:** 2

**Summary:**

The paper investigate the impact of embedding space geometry on spurious correlation in zero-shot vision-language models such CLIP. The paper aims to assess an important topic: improving robustness to spurious correlations without fine-tuning the model. The paper proposes Density-Aware Translation, which adjusts image-text similarity scores using a local density estimate derived from group reference embeddings.

**Compliance With Llm Reviewing Policy:**

Affirmed.

**Key Questions For Authors:**

1. How sensitive is DAT to the choice and quality of the reference set? For example, how does performance change if the reference samples contain mislabeled or noisy group assignments?

2. How does the method behave when spurious attributes are unknown or poorly defined? While DAT* attempts to infer attributes automatically, it is unclear how robust this process is across different domains.

**Limitations:**

yes

**Strengths And Weaknesses:**

**Strengths**

1. The paper provides an interesting perspective by connecting spurious correlations to anisotropic geometry in CLIP embedding space, where frequent patterns cluster near the mean while rare but meaningful samples lie in sparse regions. The proposed density-aware correction is conceptually simple but leverages the above geometric insight.

2. A key benefit of DAT is that it operates without fine-tuning, prompt engineering, or model parameter access. The method only requires a small reference set to estimate group densities and then rescales similarity scores accordingly. This preserves the flexibility and deployment advantages of pre-trained VLMs.

3. The authors provide a theoretical analysis of showing that cosine similarity ignores anisotropy in embedding distributions. Using a Kent distribution model, they show that DAT approximately restores missing log-likelihood terms in the decision rule and aligns scoring with Bayes-optimal discrimination under certain assumptions.

**Weaknesses**

1. The proposed method relies on group reference samples to estimate density, which may limit applicability in truly zero-shot or fully unlabeled settings. Although the DAT variant attempts to infer attributes automatically, the quality of density estimation may still depend heavily on the representativeness and size of the reference set.

2. Although the empirical results show improvements across datasets, the paper does not provide a detailed analysis of when DAT fails or provides minimal benefit, such as when spurious attributes are subtile or when the embedding density does not clearly separate groups.

3. The approach still depends on spurious attribute prompts and group prompts to construct reference sets and similarity scores. In realistic scenarios, identifying relevant spurious attributes may require domain knowledge, and incorrect or incomplete attributes definitions could affect performance.

---

> ### Author Rebuttal · Authors · 2026-03-31
>
> We sincerely appreciate your thorough review and insightful comments. Please find our responses to your questions below.
>
> **W1 (Reference labels and reference quality).**
>
> **A-W1:** The reference samples are only used to estimate geometric statistics of the embedding space. Using auxiliary samples to estimate statistics is a common practice in zero-shot calibration and debiasing methods [1].
>
> Regarding the reference set quality and size, our ablation results (Section 4, Fig. 3) show that DAT remains stable across a wide range of reference sizes, indicating that density estimation does not require large or perfectly representative reference sets.
>
> **W2 (DAT failure cases).**
>
> **A-W2:** DAT is designed to correct bias caused by density imbalance between spuriously correlated subgroups. Therefore, the largest gains are expected when the initial subgroup imbalance is stronger. This is also consistent with our empirical results from Tables 1 and 2, we observe that settings with a larger baseline gap between average and worst-group accuracy (like Waterbirds in ViT L/14 or ResNet50) often show larger improvements from DAT. This suggests that DAT is most beneficial when spurious correlations are more pronounced.
>
> **W3 (Dependence on attribute and group prompts).**
>
> **A-W3:** We agree that, like prior zero-shot debiasing methods such as TIE/TIE* [1], ROBOSHOT [2], and PerceptionCLIP [3], DAT relies on attribute and group prompts to define the relevant spurious factors. At the same time, our method is not highly sensitive to the exact prompt formulation. As discussed in Appendices D.2 and E, changing the prompt template or using different attribute descriptions has only a limited effect on performance. Table 15 further shows that DAT* is less sensitive (less than 1%) to spurious template changes compared to competing methods such as TIE*, which shows more than 5% variation in WG accuracy.
>
> **Q1 (Reference set sensitivity).**
>
> **A-Q1:** DAT is reasonably robust to noise in the reference set. As shown below, when label noise increases from 0% to 15%, the WG remains relatively stable. The average accuracy also shows only minor fluctuations. Although the GAP varies slightly at some noise levels, there is no consistent degradation trend as noise increases. These results suggest that DAT does not strongly depend on perfectly clean groups and can tolerate moderate noise in the reference set.
>
> |Noise|CLIP (ViT-B/32)|||CLIP (ViT-L/14)|||CLIP (ResNet50)|||
> |-----|:---:|:---:|:---:|:---:|:---:|:---:|:---:|:---:|:---:|
> | |WG|Avg|GAP|WG|Avg|GAP|WG|Avg|GAP|
> |0%|75.08|80.36|5.28|83.33|89.57|6.42|75.08|83.83|8.75|
> |5%|74.14|81.48|7.34|80.37|90.27|9.90|74.77|83.38|8.61|
> |10%|74.61|80.67|6.06|83.49|89.45|5.96|74.61|86.81|12.2|
> |15%|76.01|81.62|5.61|83.65|89.37|5.72|74.29|83.79|9.50|
>
> **Q2 (Robustness to unknown attributes).**
>
> **A-Q2:** We analyze the robustness of DAT* by comparing the WG identities across zero-shot classification, DAT*, and spurious attribute prediction. As shown in the table below, the WG of the spurious attribute predictor often matches the WG in zero-shot classification. This suggests that samples from the minority subgroup are more often confused with the dominant subgroup due to spurious correlations.
>
> However, DAT* changes this behavior. We observe that the WG under DAT* can differ from zero-shot, indicating that DAT* is not strongly dependent on attribute prediction accuracy. Even when the spurious attribute predictor has relatively low worst-group accuracy, DAT* still achieves substantial improvements in worst-group performance. This shows that DAT* does not require perfect attribute predictions and remains effective even when attribute inference is imperfect.
>
> | | CLIP (ViT-B/32) | | | | CLIP (ViT-L/14) | | | | CLIP (ResNet50) | | | |
> |-----------------|:-------------:|:----:|:----:|:----:|:-------------:|:----:|:----:|:----:|:-------------:|:----:|:----:|:----:|
> | | WG id | WG | Avg | GAP | WG id | WG | Avg | GAP | WG id | WG | Avg | GAP |
> | ZS | 01 | 41.37 | 68.48 | 27.11 | 01 | 31.93 | 83.72 | 51.79 | 10 | 35.36 | 80.64 | 45.28 |
> | DAT* | 10 | 64.02 | 82.33 | 18.31 | 10 | 79.75 | 87.87 | 8.12 | 10 | 63.71 | 82.65 | 18.94 |
> | Attr. Pred. | 01 | 63.19 | 77.73 | 14.54 | 01 | 49.67 | 75.05 | 25.38 | 10 | 51.49 | 67.92 | 16.43 |
>
> [1] Lu, Shenyu, Junyi Chai, and Xiaoqian Wang. "Mitigating Spurious Correlations in Zero-Shot Multimodal Models." International Conference on Learning Representations (ICLR), 2025.
>
> [2] Adila, Dyah, et al. "Zero-Shot Robustification of Zero-Shot Models." International Conference on Learning Representations (ICLR), 2024.
>
> [3] An, Bang, et al. "PerceptionCLIP: Visual Classification by Inferring and Conditioning on Contexts." International Conference on Learning Representations (ICLR), 2024.

---

> > ### Author Rebuttal · Reviewer_4Vkd · 2026-04-04
> >
> > The rebuttal introduces additional experiments on reference set noise, showing relatively stable performance up to 15% label corruption, though the evaluation is limited to specific settings. The discussion of failure cases mainly links improvements to scenarios with stronger subgroup imbalance, but does not include controlled experiments demonstrating behavior when such imbalance is weak or absent. Prompt sensitivity experiments indicate limited variation under different templates, but the method still requires predefined or inferred attributes. The analysis of DAT* suggests improvements even with imperfect attribute prediction, though the evidence is indirect and evaluated on a limited set of scenarios.

---

> > > ### Author Response · Authors · 2026-04-05
> > >
> > > Thanks for your time and for carefully considering our rebuttal.
> > >
> > > **Settings for noise:**
> > >
> > > Due to the character limit, our initial analysis considered uniform reference noise on Waterbirds across different backbones. Here, we additionally evaluate an important setting where noise is applied specifically to the minority/conflicting subgroups (01 and 10). As shown below, performance remains stable as noise increases from 0% to 15%. WG accuracy varies only slightly and does not exhibit consistent degradation. Minor WG improvements reflect mild stabilisation effects in density estimation. Overall, the results confirm DAT’s robustness to structured reference noise.
> > >
> > > |Noise|CLIP (ViT-B/32)|||CLIP (ViT-L/14)|||CLIP (ResNet50)|||
> > > |-----|---------------|---|---|---------------|---|---|---------------|---|---|
> > > |     |WG|Avg|GAP|WG|Avg|GAP|WG|Avg|GAP|
> > > |0%|75.08|80.36|5.28|83.33|89.57|6.42|75.08|83.83|8.75|
> > > |5%|76.49|81.03|4.54|84.11|89.51|5.40|76.48|84.78|8.30|
> > > |10%|78.04|81.67|3.63|83.80|89.94|6.14|76.58|83.33|6.75|
> > > |15%|78.66|81.55|2.89|85.14|87.87|2.73|78.49|83.45|4.96|
> > >
> > > Also, we observe similar stability on the CelebA dataset, further confirming that DAT’s robustness to reference noise generalises beyond Waterbirds:
> > >
> > > |Noise|CLIP (ViT-B/32)|||CLIP (ViT-L/14)|||CLIP (ResNet50)|||
> > > |-----|---------------|---|---|---------------|---|---|---------------|---|---|
> > > |     |WG|Avg|GAP|WG|Avg|GAP|WG|Avg|GAP|
> > > |0%|78.53|87.09|8.56|85.35|86.54|1.19|80.79|87.09|6.30|
> > > |5%|79.09|86.76|7.67|84.86|86.47|1.61|82.49|85.85|3.36|
> > > |10%|79.09|86.64|7.55|84.84|86.46|1.62|81.92|86.31|4.39|
> > > |15%|79.09|86.60|7.51|84.84|86.46|1.62|84.02|86.11|2.09|
> > >
> > > **Weak imbalance:**
> > >
> > > Unlike Waterbirds, which exhibits strong imbalance, CelebA provides a setting with much weaker subgroup disparity, as reflected by the substantially smaller baseline gap (e.g., 5.38 vs 27.11 for ViT-B/32). In this setting, DAT still improves robustness but with naturally smaller gains, since density differences between subgroups are less pronounced. This behaviour is consistent with the mechanism of DAT, when subgroup density contrast is weaker, the density-based correction has less structure to exploit, and improvements are therefore more moderate.
> > >
> > > | Method | CLIP (ViT-B/32) |  | CLIP (ViT-L/14) |  | CLIP (ResNet50) |  |
> > > |--------|-----------------|--|-----------------|--|-----------------|--|
> > > |        | GAP (Waterbirds) | GAP (CelebA) | GAP (Waterbirds) | GAP (CelebA) | GAP (Waterbirds) | GAP (CelebA) |
> > > | ZS | 27.11 | 5.38 | 51.79 | 7.85 | 45.28 | 11.89 |
> > > | Group Prompt | 23.33 | 5.48 | 45.68 | 8.92 | 21.12 | 8.89 |
> > > | DAT | 5.28 | 8.56 | 6.42 | 1.19 | 8.75 | 6.30 |
> > >
> > > **Prompt sensitivity and inferred attributes:**
> > >
> > > As discussed, DAT follows the standard assumptions used in prior zero-shot debiasing works [1–3] and is not highly sensitive to prompt design and attribute description. We further note that Tables 4 and 11–12 (Appendix D.2) confirm consistent behaviour across semantically related variants of spurious attributes on Waterbirds and CelebA. Appendix E (Tables 15–16) also demonstrates stability under template changes and different levels of attribute description specificity.
> > >
> > > **DAT\* and attribute prediction:**
> > >
> > > In the Waterbirds analysis above, we showed that DAT* can improve robustness even with imperfect attribute prediction. Here, CelebA provides a complementary scenario with a weaker imbalance and strong attribute prediction accuracy. We observe that DAT* remains effective in this setting as well. Together, these results suggest that DAT* is not highly sensitive to attribute prediction quality and performs reliably across different attribute conditions.
> > >
> > > | Method | CLIP (ViT-B/32) |  |  | CLIP (ViT-L/14) |  |  | CLIP (ResNet50) |  |  |
> > > |--------|-----------------|--|--|-----------------|--|--|-----------------|--|--|
> > > |        | WG | Avg | GAP | WG | Avg | GAP | WG | Avg | GAP |
> > > | ZS     | 78.89 | 84.27 | 5.38 | 73.35 | 81.20 | 7.85 | 69.69 | 81.58 | 11.89 |
> > > | DAT*   | 78.53 | 87.11 | 8.58 | 84.93 | 86.54 | 1.19 | 78.53 | 88.29 | 9.76 |
> > > | Attr. Pred.	 | 90.39 | 96.81 | 6.42 | 89.83 | 96.77 | 6.94 | 89.83 | 96.74 | 6.91 |
> > >
> > > We would be happy to provide additional analysis if there are specific scenarios the reviewer would like us to examine.

---

### Decision · Program_Chairs · 2026-04-30

**Decision:**

Accept (regular)

**Comment:**

This paper introduces Density-Aware Translation (DAT), a training-free method to mitigate spurious correlations in zero-shot vision-language models. Reviewers consistently recognize the method's technical soundness, novelty, simplicity and its theoretical grounding in the Kent distribution. The authors provided an outstanding rebuttal that addressed all initial concerns. The reviewers reached a positive consensus (3 Weak Accept and 1 Accept) after the rebuttal.